# A glycine-rich PE_PGRS protein governs mycobacterial actin-based motility

Norbert S. Hill[1✉] & Matthew D. Welch [1✉]

Many key insights into actin regulation have been derived through examining how microbial pathogens intercept the actin cytoskeleton during infection. *Mycobacterium marinum*, a close relative of the human pathogen *Mycobacterium tuberculosis*, polymerizes host actin at the bacterial surface to drive intracellular movement and cell-to-cell spread during infection. However, the mycobacterial factor that commandeers actin polymerization has remained elusive. Here, we report the identification and characterization of the *M. marinum* actin-based motility factor designated mycobacterial intracellular rockets A (MirA), which is a member of the glycine-rich PE_PGRS protein family. MirA contains an amphipathic helix to anchor into the mycobacterial outer membrane and, surprisingly, also the surface of host lipid droplet organelles. MirA directly binds to and activates the host protein N-WASP to stimulate actin polymerization through the Arp2/3 complex, directing both bacterial and lipid droplet actin-based motility. MirA is dissimilar to known N-WASP activating ligands and may represent a new class of microbial and host actin regulator. Additionally, the MirA-N-WASP interaction represents a model to understand how the enigmatic PE_PGRS proteins contribute to mycobacterial pathogenesis.

[1] Department of Molecular and Cell Biology, University of California, Berkeley, CA, USA. ✉email: bisco@berkeley.edu; welch@berkeley.edu

Elucidating how microbial pathogens manipulate host cell pathways and structures during infection has illuminated many aspects of host cell biology. A frequent target of infectious microbes is the actin cytoskeleton, and microbes have developed an array of mechanisms to commandeer actin polymerization for entry, establishing a replicative niche, and egress[1–4]. A diverse set of pathogens have evolved the ability to polymerize actin directly against the microbial surface to power actin-based motility. This propels the microbe within and between host cells, driving tissue dissemination while also allowing continued access to cytosolic nutrients and avoidance of host immune responses[5]. *Mycobacterium marinum*, an aquatic pathogen that occasionally infects humans, and a model organism to study the significant human pathogen *Mycobacterium tuberculosis*[6, 7], is one such pathogen that undergoes actin-based motility[8–10]. To polymerize actin, *M. marinum* recruits the host cell nucleation promoting factors WASP and/or N-WASP, which promote actin filament nucleation by binding to and activating the host Arp2/3 complex. However, the bacterial factor governing actin-based motility in *M. marinum* has remained elusive, raising the possibility that *M. marinum* has evolved a distinctive WASP/N-WASP activation strategy. In this work, we identify a glycine-rich PE_PGRS family protein as the *M. marinum* actin-based motility factor, which directly binds and activates host N-WASP to stimulate actin polymerization against the bacterial surface.

## Results

**A screen identifies *MMAR_3581* as critical for *M. marinum* cell-to-cell spread.** To identify the *M. marinum* actin-based motility factor and other genes involved in cell-to-cell spread, we assessed >35,000 individual *M. marinum* transposon insertion mutants by fluorescence microscopy for defects in cell-to-cell spread through a confluent monolayer of U2OS host cells. Within the subset of genes that were crucial for cell-to-cell spread without causing a marked growth defect, we focused on *MMAR_3581* (*MMAR_RS17840*) (Fig. 1a) because its product is known to be secreted across the bacterial envelope[11]. *M. marinum* is the only mycobacterial species to encode *MMAR_3581* except for a truncated form encoded by the closely related pathogen *Mycobacterium ulcerans*. Intriguingly, the genomic region adjacent to *MMAR_3581* exhibits multiple features of horizontal gene acquisition (Supplementary Fig. 1), suggesting that *M. marinum* may have acquired *MMAR_3581* from another mycobacterial species. MMAR_3581 is a member of the PE_PGRS protein family, which are translocated through the ESX-5 type VII secretion system[12]. Consistent with an ESX-5 substrate being involved in *M. marinum* actin-based motility, two separate transposon insertions into *eccA₅*, a cytosolic component of ESX-5, were also identified in our screen as having significantly reduced bacterial cell-to-cell spread (Fig. 1b).

We generated a clean *MMAR_3581* deletion strain and assessed for its ability to spread through a monolayer of U2OS cells by measuring the number of cells infected per infectious focus. Wild-type *M. marinum* spread efficiently between host cells over a time course of 72 h, however, the *MMAR_3581* deletion strain failed to spread (Fig. 1c, d). The defective spread was not due to diminished replication, as the Δ*MMAR_3581* bacteria had indistinguishable growth compared to wild type in broth and during infection of either U2OS cells or mouse bone marrow-derived macrophages (BMDMs) (Fig. 1e and Supplementary Fig. 2a, b). Further, complementing the Δ*MMAR_3581* strain with a chromosomally integrated copy of *MMAR_3581* driven by its native promoter restored cell-to-cell spread (Fig. 1c, d). Thus, MMAR_3581, an ESX-5 translocated protein, is necessary for *M. marinum* cell-to-cell spread.

**MMAR_3581 is necessary to recruit host N-WASP for actin-based motility.** We next tested if the Δ*MMAR_3581* mutant was defective for actin-based motility or a different step in cell-to-cell spread (e.g., vacuolar escape or protrusion formation). U2OS cells were infected with wild type, Δ*MMAR_3581*, or *MMAR_3581*-complemented bacteria, then assessed for association with actin clouds, actin tails, and N-WASP over the course of infection. Wild-type *M. marinum* and the complemented strain recruited N-WASP and generated actin tails by ~12 h post infection (hpi) with a frequency that steadily increased to 36% of the bacterial population by 72 hpi (Fig. 2a, b). Conversely, Δ*MMAR_3581* bacteria were unable to recruit N-WASP and failed to polymerize actin throughout infection. Further, Δ*MMAR_3581* bacteria also failed to undergo actin-based motility in BMDMs (which express both WASP and N-WASP[9], whereas U2OS cells only express N-WASP), suggesting that *MMAR_3581* is necessary for recruiting both WASP and N-WASP (Supplementary Fig. 3a, b). Over-expressing MMAR_3581 >9-fold prompted an earlier

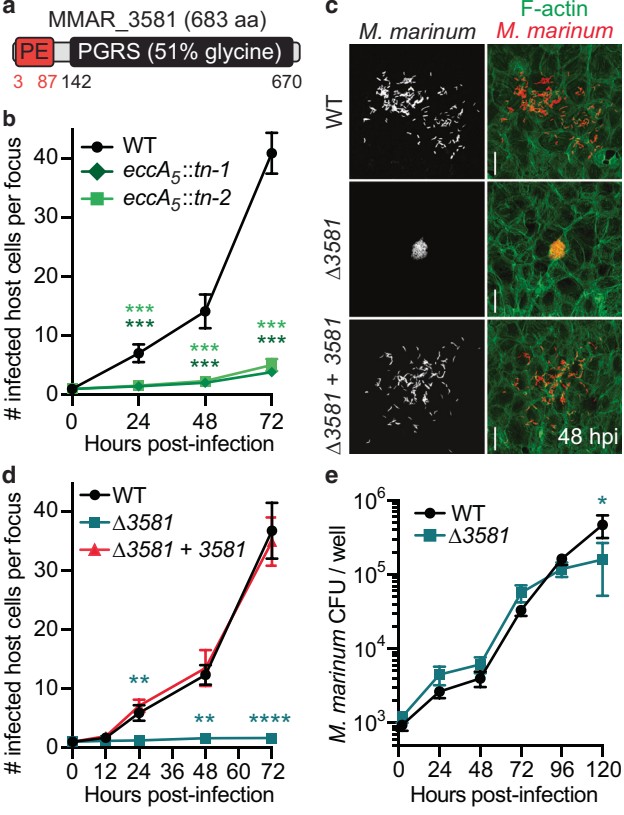

**Fig. 1 *MMAR_3581*, an ESX-5-secreted substrate, is necessary for *M. marinum* cell-to-cell spread. a** Domain structure of the PE_PGRS protein MMAR_3581. **b** Time course graph indicating the number of host U2OS cells in an infectious focus during infection of *M. marinum* wild type or *eccA₅* transposon inserted strains. **c** Micrographs depicting bacterial cell-to-cell spread within a host cell monolayer in a single infectious focus with specified *M. marinum* (red; tdTomato) strains in U2OS host cells (green; Alexa 488 phalloidin) at 48 hpi. Scale bar is 10 μm and images are representative of three independent experiments. **d** Time course graph indicating the number of host U2OS cells per infectious focus during infection of indicated *M. marinum* strains. **e** Growth curve of indicated *M. marinum* strains during infection of U2OS host cells. For **b**, **d**, **e** data is mean ± SD; *n* = 3 independent experiments. Data in **b**, **d** used a one-way ANOVA with post-hoc Tukey test was where each group was compared to wild type and data in **e** used an unpaired two-tailed *t* test; \**p* < 0.05, \*\**p* < 0.01, \*\*\**p* < 0.001, and \*\*\*\**p* < 0.0001. Source data are provided as a Source data file.

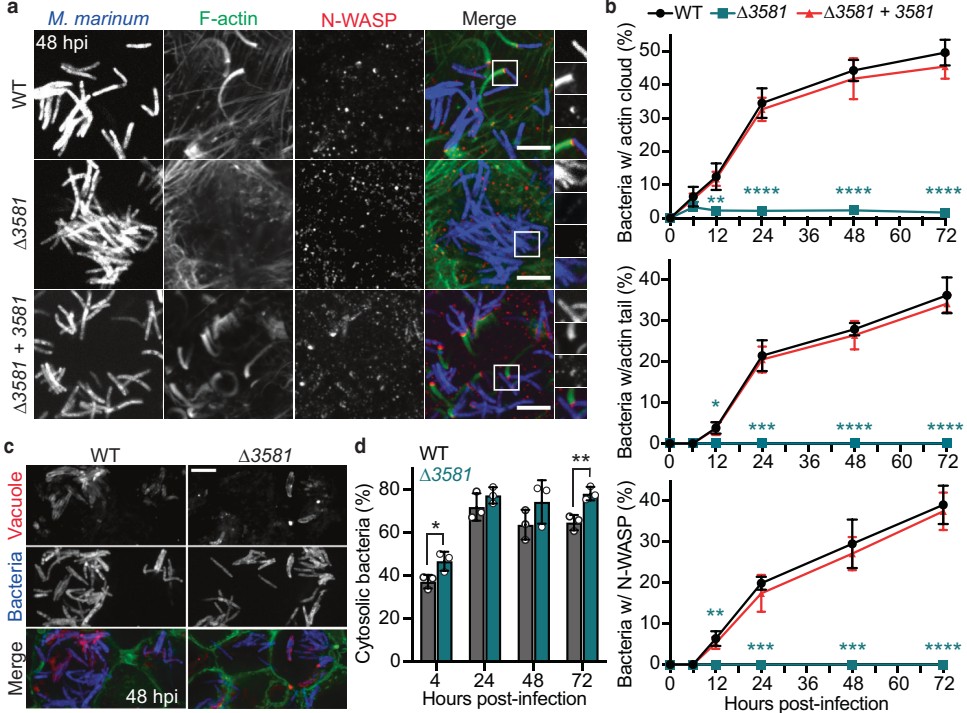

**Fig. 2 Δ*MMAR_3581* bacteria escape into the host cytosol but are unable to recruit N-WASP to the bacterial surface. a** Micrographs depicting the ability of indicated *M. marinum* (blue; EBFP2) strains to recruit N-WASP (red; anti-N-WASP) and stimulate actin polymerization (green; Alexa 488 phalloidin) in U2OS host cells at 48 hpi. Scale bar is 5 µm and images are representative of three independent experiments. **b** Time course graphs of indicated *M. marinum* stains colocalized with either actin clouds, actin tails, or N-WASP. **c** Representative micrographs of indicated *M. marinum* (blue; EBFP2) strains enveloped by a vacuole (red; CM-DiI) in Raw 264.7 cells (green; Alexa 488 phalloidin) at 48 hpi to examine cytoplasmic access. Scale bar is 3 µm. **d** Quantification of cytoplasmic access for indicated *M. marinum* strains during infection of Raw 264.7 cells at indicated time points. Data in **b**, **d** are mean ± SD; $n = 3$ independent experiments. Data in **b** used a one-way ANOVA with post hoc Tukey test where each group was compared to wild type and data in **d** used an unpaired two-tailed *t* test; *$p < 0.05$, **$p < 0.01$, ***$p < 0.001$, and ****$p < 0.0001$. Source data are provided as a Source data file.

timing of actin-based motility (Supplementary Fig. 4a); however, the increased MMAR_3581 levels engendered bacterial filamentation that reduced actin-based motility and spread at later time points (Supplementary Fig. 4a-d). Further, the Δ*MMAR_3581* bacteria were not deficient in actin-based motility due to a defect in escaping from the vacuole into the host cytoplasm. Δ*MMAR_3581* bacteria were observed in the host cytosol with same or higher frequency as wild type when either staining the vacuolar membrane (Fig. 2c, d) or by transmission electron microscopy (Supplementary Fig. 5a, b). In addition, Δ*MMAR_3581* bacteria had a similar frequency of polyubiquitin colocalization compared with wild type, which serves as an indirect marker of vacuolar integrity (Supplementary Fig. 5c, d)[13, 14]. This also indicates that MMAR_3581 may not play a role in autophagy evasion like other bacterial actin-based motility effector proteins[15–17]. Altogether, these data demonstrate MMAR_3581 is necessary to recruit WASP/N-WASP to stimulate actin-based motility during infection. As such, we have designated this gene as *mycobacterial intracellular rocketing A* (*mirA*).

**MirA localizes at sites of actin polymerization on the bacterial surface.** Previous work demonstrated that *M. marinum* only initiates actin-based motility in response to the host cell environment[8]. To examine MirA expression and secretion, MirA in the bacterial outer envelope was extracted using the detergent Genapol X-080 from *M. marinum* grown in either broth or RAW 264.7 macrophage cells, then probed using an antibody raised against purified MirA. MirA was only observed during infection

of host cells and coincided with the timing of actin-based motility (Fig. 3a). We next investigated the localization of MirA during infection by immunofluorescence microscopy. MirA was observed at the bacterial surface starting at 8 hpi, and the frequency of MirA surface localization increased over time (Fig. 3b–d). At 72 hpi, the percent of MirA-positive bacteria with surface-associated N-WASP or an actin tail was 71% or 63%, respectively. We observed that the distribution of MirA on the bacterial surface dynamically transitioned over time from a mostly punctate distribution at early time points (8 hpi) to a single polar distribution at later time points (Fig. 3e, f). Together, these data demonstrate that MirA is expressed and secreted in response to infection, and over time it localizes primarily to a single bacterial pole at which N-WASP is recruited and actin is polymerized.

Next, we tested whether ectopically supplied MirA could restore *M. marinum* actin-based motility of Δ*mirA* bacteria. Host cells were transfected to express full-length MirA or a variant lacking the PE domain (MirA$^{ΔPE}$; PE domain, residues 1–87) (Supplementary Fig. 6b), which approximates the proteolytically processed secreted form (Supplementary Fig. 6c)[11, 18]. Host cells were subsequently infected with Δ*mirA* bacteria and stained for MirA and F-actin at 48 hpi. Ectopically expressed MirA and MirA$^{ΔPE}$ both localized to the bacterial surface and restored actin polymerization (Fig. 3g and Supplementary Fig. 7a, b). In some cases, MirA concentrated at a single bacterial pole and promoted the assembly of an actin comet tail. This indicates that MirA, without its PE domain, can bind to the bacterial surface and has the ability to complement Δ*mirA* bacteria in trans.

                    3

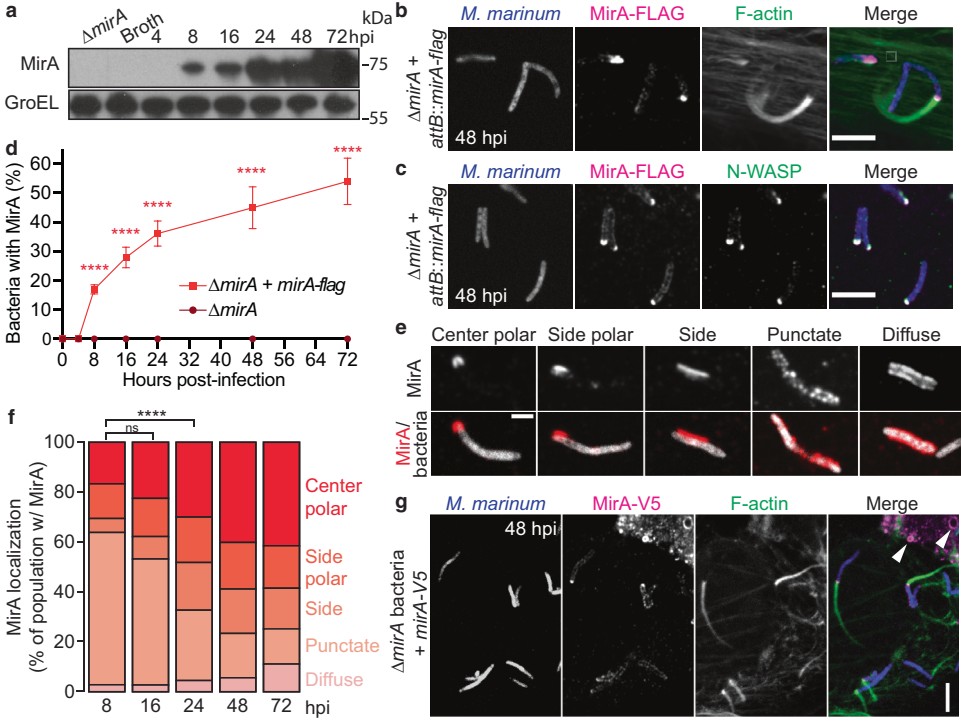

**Fig. 3 MirA is secreted in response to infection and colocalizes with N-WASP and F-actin. a** Immunoblot of MirA levels on the bacterial surface (Genapol X-080 supernatant) in broth or during infection of RAW 264.7 cells. GroEL2 is shown as the loading control. Endogenously expressed MirA-FLAG (magenta; anti-Flag) with **b** F-actin (green; Alexa 488 phalloidin) or **c** N-WASP (green; anti-N-WASP) colocalization at the *M. marinum* (blue; EBFP2) surface during infection of U2OS cells at 48 hpi. Scale bar is 5 µm. **d** Quantification of indicated *M. marinum* strains with MirA surface localization during infection of U2OS cells. **e** Representative micrographs depicting the five categories of MirA-FLAG localization on the bacterial surface during infection of U2OS cells at 48 hpi. Scale bar is 1 µm. **f** Time course of MirA-FLAG localization on the bacterial surface during infection of U2OS cells. **g** Ectopically expressed MirA-V5 (magenta; anti-V5) localization with Δ*mirA* bacteria (blue; tdTomato) and F-actin (green; Phalloidin-iFluor 405) in U2OS cells 52 hpt and 48 hpi. Scale bar is 3 µm and white arrows indicate transfected MirA localizing to eukaryotic lipid droplet organelles. Data in **d** are the mean ± SD; $n = 3$ independent replicates and an unpaired two-tailed $t$ test was used to assess significance. The data in **f** is the aggregate of 3 independent experiments and a chi-square test is used; ns = non-significant and ****$p < 0.0001$. Data shown in **a**, **g** are representative of three independent experiments. Source data are provided as a Source data file.

**Ectopically expressed MirA localizes to and propels host lipid droplets**. Curiously, in non-infected eukaryotic cells, ectopically expressed MirA had a distinctive localization to spherical structures (Fig. 3g, white arrows). We determined the MirA-coated spheres to be lipid droplets (Fig. 4a), eukaryotic organelles central to lipid and energy homeostasis, and hubs for signaling and host defense[19, 20]. Expression of MirA did not affect lipid droplet size or number (Supplementary Fig. 8a, b). However, in cells expressing MirA we observed robust actin polymerization at the lipid droplet surface and the formation of actin comet tails. To explore whether MirA was inducing actin-based movement of lipid droplets, MirA was expressed in a U2OS cell line stably expressing F-tractin-mCherry (a fluorescent marker of F-actin) and observed by live-cell imaging. Remarkably, MirA-coated lipid droplets exhibited sustained actin-based motility (Fig. 4b and Supplementary Movies 1–3). The movement velocity (µm/min) and efficiency (final displacement/total distance traveled) were compared between *M. marinum* and lipid droplets. Lipid droplets moved a third slower (mean 13 versus 18 µm/min), but in a more efficiently path (mean 0.7 versus 0.5) relative to *M. marinum* (Fig. 4c).

To determine if the molecular mechanism of actin-based motility is similar between MirA-coated lipid droplets and *M. marinum*, we stained for N-WASP and the Arp2/3 complex in *mirA* cells transfected. We observed that both N-WASP and the Arp2/3 complex localize to lipid droplets, though only in cells with MirA expression (Fig. 4d, e). Next, to assess whether MirA-

coated lipid droplets could spread into a neighboring cell like *M. marinum*, we used a mixing experiment in which cells expressing a plasma membrane marker (RFP-CAAX) were transfected to express MirA, then co-plated with unmarked, non-transfected cells and assessed for lipid droplet transfer after 36 h. Remarkably, we observed 5–10% of MirA-coated lipid droplets were found in protrusions of the plasma membrane or in recipient neighboring cells (Fig. 4f and Supplementary Fig. 8c). Together, these data indicate that MirA is sufficient to induce actin-based motility and cell-to-cell spread of bacteria and lipid droplet organelles.

**MirA uses an amphipathic helix to localize to the surface of lipid droplets and *M. marinum*.** To understand how MirA localizes to the surface of lipid droplets and *M. marinum*, we examined MirA for potential membrane-targeting sequences. Using HELIQUEST, an online amphipathic helix prediction algorithm[21], a candidate amphipathic helix was identified in the linker region between the PE domain and the PGRS domain (AH domain: residues 102–119) (Fig. 5a). When the predicted amphipathic helix was fused to tdTomato (MirA$^{AH}$-tdTomato) and expressed in U2OS cells, MirA$^{AH}$-tdTomato localized to both the surface of lipid droplet organelles and *M. marinum* (Fig. 5b, c, g). Conversely, ectopically expressing an amphipathic helix deletion mutant (MirA$^{ΔAH}$) failed to localize to the surface of either lipid droplets or Δ*mirA* bacteria (Fig. 5d, e, g, Supplementary Figs. 6b, and 7a). Likewise, in spite of normal expression and secretion (Supplementary Fig. 6a, c), we were unable to detect

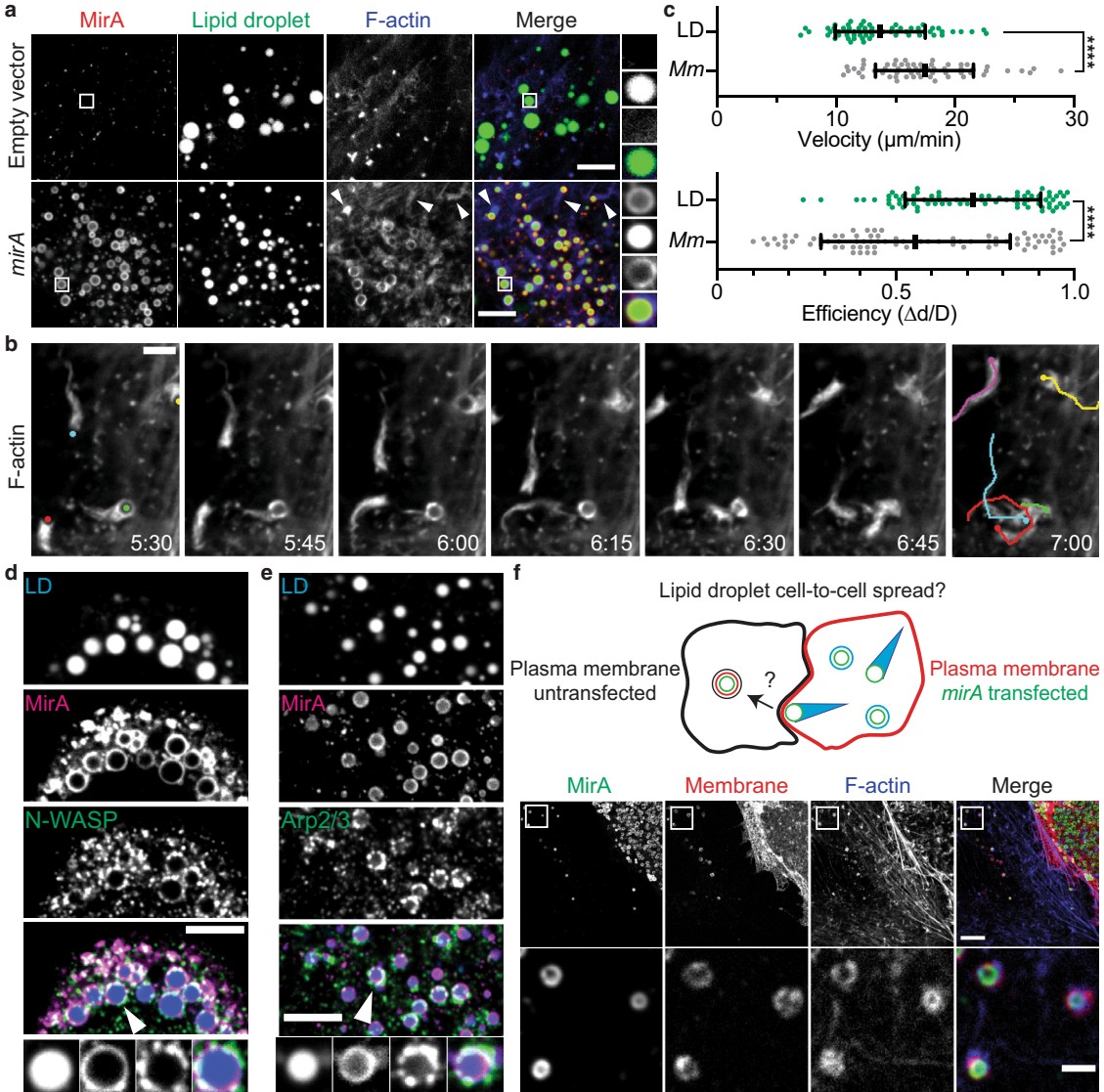

**Fig. 4 MirA localizes to eukaryotic lipid droplets and mimics *M. marinum* actin-based motility. a** Ectopically expressed MirA (red; anti-MirA) in U2OS cells localizes to lipid droplet organelles (green; BODIPY 493/503) and induces actin polymerization (blue; Phalloidin-iFluor 405). Arrows indicate actin comet tails. Scale bar is 3 μm. **b** Live-imaging snapshots of *mirA^{ΔPE}* transfected U2OS cells expressing the F-actin marker F-tractin-mCherry. Scale bar is 3 μm. Accompanies Supplementary Movie 2. **c** Velocity of movement and efficiency of movement measurements of *M. marinum* (*Mm*; gray) versus lipid droplets (LD; green). The data are a combination of ≥20 measurements over three independent biological experiments and the bar indicates the mean ± SD. Statistical analysis used a two-tailed Mann–Whitney U test; ****$p < 0.0001$. Transfected MirA (red; anti-MirA) localizes with **d** N-WASP (green; anti-N-WASP) or **e** Arp2/3 (green; anti-P34) at the surface of lipid droplet organelles (blue; BODIPY 493/503). Scale bar is 3 μm and the white arrow in merge panel indicates highlighted example. **f** MirA- (green; anti-MirA) coated lipid droplets spread from donor *mirA*-transfected A549 cells expressing a plasma membrane marker (red; RFP-CAAX) to unmarked and untransfected recipient A549 cells. Bottom panel corresponds to boxed region. Scale bar is 5 μm (upper panel) and 2 μm (lower panel). Data shown in **a**, **b**, **d**–**f** are representative of at least three independent experiments. Source data are provided as a Source data file.

endogenously expressed MirA^{ΔAH} or actin filaments at the bacterial surface by immunofluorescence microscopy (Fig. 5f, g). Instead, we detected both exogenously and endogenously expressed MirA^{ΔAH} localizing in punctate foci in the host cytoplasm (Fig. 5d–g). These data indicate that the MirA amphipathic helix is both necessary and sufficient for localization to the bacterial and lipid droplet surfaces. In addition, we probed for MirA by subcellular fractionation and detected wild-type levels of MirA^{ΔAH} in the Genapol X-080 supernatant (Supplementary Fig. 6c). This suggests MirA^{ΔAH} localizes and is retained in the bacterial outer envelope. Since we do not observe surface-associated MirA^{ΔAH} or actin polymerization by microscopy, this

may represent a MirA fraction that is not fully secreted to the surface or correctly oriented in the outer leaflet to recruit the actin machinery. Alternatively, this data could suggest MirA has additional surface localization mechanisms beyond the amphipathic helix.

Bioinformatic analysis of both *M. tuberculosis* and *M. marinum* PE_PGRS proteins, which are largely thought to localize to the bacterial surface[11, 22–27], revealed that many contain a candidate amphipathic helix in their linker region (Fig. 5h, Supplementary Fig. 9a, and Supplementary Data 1). We found another *M. marinum* PE_PGRS protein, MMAR_2645, also localizes to lipid droplets when ectopically expressed in eukaryotic cells

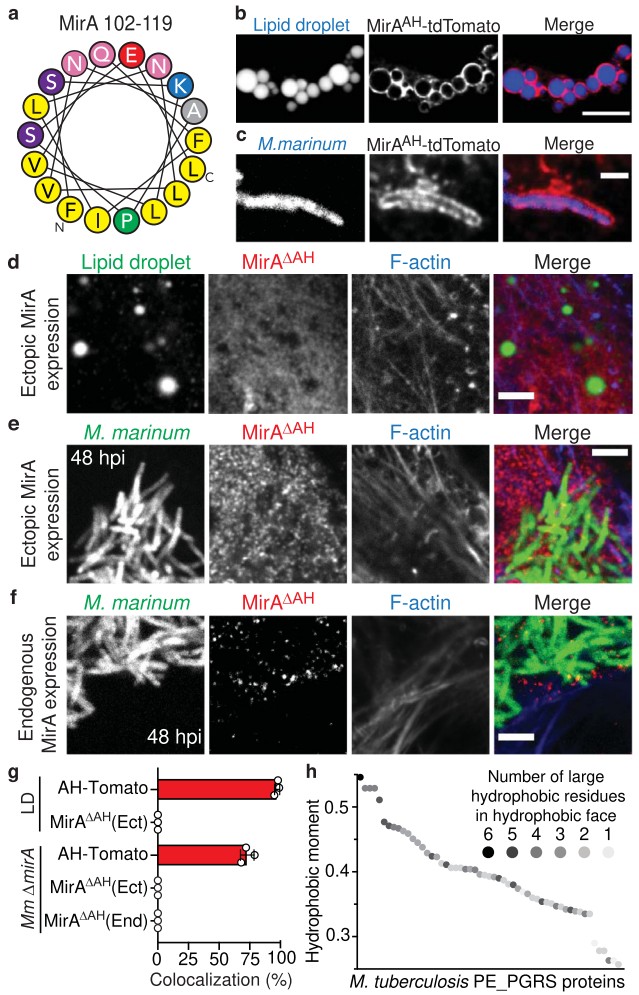

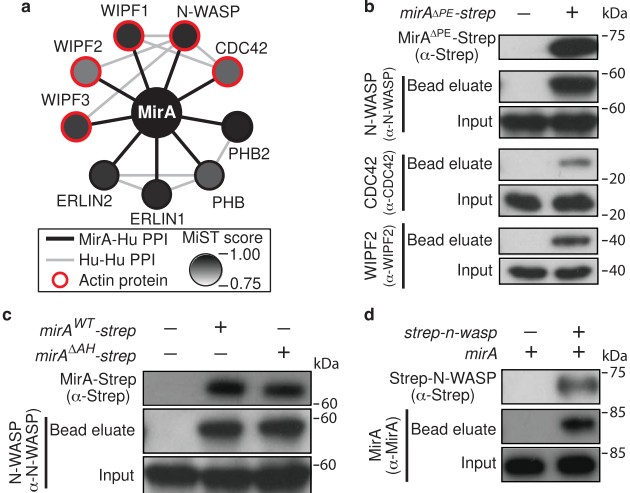

**Fig. 6 MirA interacts with N-WASP and N-WASP-interacting proteins involved in actin regulation. a** A schematic of the MirA-host interactome determined through an AP-MS approach in HEK293 cells detecting protein-protein interactions (PPI) determined using the MiST algorithm[28]. Black lines indicate MirA-human protein interactions, gray lines indicate human-human interactions (via STRING[72]), and actin cytoskeletal proteins are circled in red. **b** Representative immunoblots of MirA-Strep affinity-purified lysates blotted against actin cytoskeletal proteins (N-WASP, CDC42, and WIPF2) identified as MirA interacting factors. **c** Representative immunoblots of the MirA-Strep amphipathic helix mutant affinity-purified lysates against N-WASP. **d** Representative immunoblots of the Strep-N-WASP affinity-purified lysates against MirA. Data shown in **b**–**d** are representative of three independent experiments. Source data are provided as a Source data file.

**Fig. 5 MirA encodes an amphipathic helix that inserts into the surface of lipid droplets and *M. marinum*. a** A putative amphipathic helix in MirA residues 102–119. **b** The MirA amphipathic helix alone fused to tdTomato ectopically expressed in U2OS cells localizes to the surface of both **b** eukaryotic lipid droplet organelles (blue; BODIPY 493/503) and **c** intracellular *M. marinum* (blue; EBFP2). Scale bars are 2 μm and 1 μm, respectively. Ectopically expressed MirA$^{\Delta AH}$ (magenta; anti-V5) is unable to localize to the surface of **d** lipid droplets (green; BODIPY 493/503) or **e** Δ*mirA M. marinum* (green; tdTomato) to promote actin polymerization (blue; Phalloidin-iFluor 405). Scale bars are both 3 μm. **f** An endogenously expressed amphipathic helix MirA$^{\Delta AH}$ mutant (magenta; anti-FLAG) is unable to localize to the surface of Δ*mirA M. marinum* (blue; EBFP2) to stimulate actin polymerization (green; Alexa 488 phalloidin). Scale bar is 3 μm. **g** Percent colocalization of MirA$^{AH}$-tdTomato (AH-Tomato) or MirA$^{\Delta AH}$ expressed ectopically (Ect) or endogenously (End) at the surface of either lipid droplets or *M. marinum* of examples shown in (**b**–**f**). Data are mean ± SD; *n* = 3 independent experiments. **h** PE_PGRS proteins from *M. tuberculosis* strain H37Rv assessed for candidate amphipathic helices in the linker region between their PE and PGRS domains. The best hydrophobic moment score is shown for each PE_PGRS protein and the number of large hydrophobic residues within the hydrophobic face, a predictive factor of amphipathic helix insertion into phospholipid monolayers[74], is shown in greyscale (*inset*). Related to Supplementary Data 1a. Source data are provided as a Source data file.

(Supplementary Fig. 9b, c), although this protein did not elicit actin polymerization. These data demonstrate that amphipathic helices are likely a common feature of mycobacterial PE_PGRS proteins that may play a role in protein localization.

**A non-biased approach identifies N-WASP as a MirA interacting protein**. We next leveraged the ectopic expression system to determine if MirA interacts with N-WASP and to explore other interactions with host proteins on lipid droplets using a non-biased affinity purification-mass spectrometry (AP-MS) approach. MirA$^{\Delta PE}$ C-terminally fused with a One-STrEP-tag (MirA$^{\Delta PE}$-Strep) was expressed in HEK293 cells, then purified by Strep-Tactin affinity chromatography. Differences in the mass spectrometric profile of proteins purified from cells transfected with *mirA$^{\Delta PE}$-strep* versus an empty plasmid control were assessed by the mass spectrometry interaction statistics algorithm (MiST), a computational tool for scoring protein–protein interactions[28]. MirA-Strep eluates were enriched for N-WASP, three members of the WASP-interacting protein family (WIPF1, WIPF2, and WIPF3), and the canonical N-WASP activator CDC42 (Fig. 6a and Supplementary Data 2)[29, 30]. This approach also identified an interaction with a putative Prohibitin-Prohibitin2-Erlin1-Erlin2 complex; however, these proteins are not known to be involved in actin cytoskeletal dynamics and were not examined further. N-WASP, CDC42, and WIPF2 were additionally detected in immunoprecipitates from cells expressing MirA-Strep by immunoblotting (Fig. 6b). In addition, the MirA$^{\Delta AH}$ mutant was still able to pull down N-WASP, demonstrating that MirA could mediate this interaction in cells without being affixed to the surface of lipid droplets (Fig. 6c). An inverse approach using an N-terminal One-STrEP tagged N-WASP as an affinity matrix pulled down MirA when they were co-expressed (Fig. 6d). Together, these data demonstrate that the actin nucleation promoting factor N-WASP and N-WASP-binding proteins are within the MirA interactome.

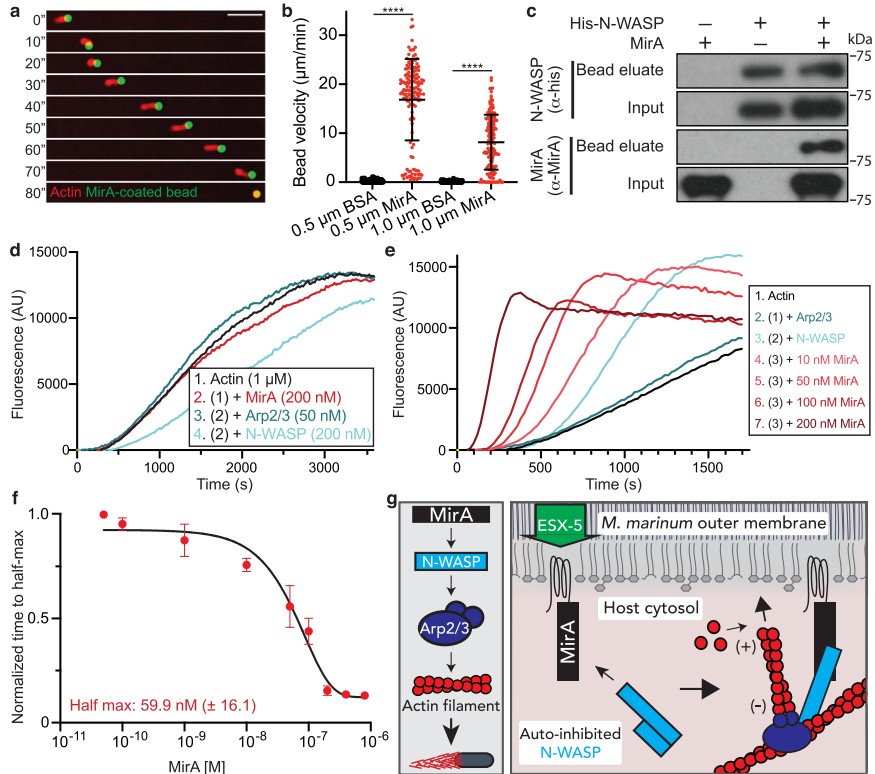

**Fig. 7 MirA directly activates N-WASP to stimulate actin polymerization through the Arp2/3 complex. a** Time-lapse micrographs of a 1 μm fluorescent polystyrene bead (green) coated with native MirA in *Xenopus laevis* egg extract with rhodamine-actin (red). Scale bar is 5 μm. Accompanies Supplementary Movie 4. **b** Average velocity (μm/min) of either 0.5 μm or 1 μm beads coated in either BSA (black) or MirA (red) in *Xenopus laevis* egg extract. Accompanies Supplementary Movie 5. The data are a combination of ≥45 measurements over three independent experiments and the bar indicates the mean ± SD. Statistical analysis used a two-tailed Mann–Whitney U test; ****$p < 0.0001$. **c** A pulldown assay using purified 6xHis-N-WASP as bait and purified MirA as prey probed with anti-His or anti-MirA by immunoblot and is representative of three independent experiments. **d** Pyrene actin (1 μM, 10% labeled) polymerization reactions with either MirA (200 nM), MirA and the Arp2/3 complex (50 nM), or MirA and N-WASP (200 nM). **e** Pyrene actin (1 μM, 10% labeled) polymerization reactions with the Arp2/3 complex (50 nM), N-WASP (200 nM), and increasing MirA concentrations. **f** Time to half-maximum fluorescence of the polymerization curves normalized to N-WASP and the Arp2/3 complex without MirA. Data are mean ± SD; $n = 3$, and the data set was fit to estimate the concentration at which half-maximum activity is observed. **g** A model of *M. marinum* actin-based motility whereby MirA is translocated by ESX-5, inserts into the outer membrane using an amphipathic helix, then recruits and directly activates host N-WASP to stimulate actin polymerization against the bacterial surface for propulsion. Source data are provided as a Source data file.

**MirA directly activates N-WASP to stimulate actin filament nucleation.** To begin to assess the biochemical activity of MirA in actin-based motility, we purified recombinant MirA protein, bound it to the surface of fluorescent polystyrene beads, and incubated MirA-coated beads in a *Xenopus laevis* egg extract supplemented with rhodamine-labeled actin. The MirA-coated beads initiated actin-based motility and moved at a similar velocity to *M. marinum* or MirA-coated lipid droplets (0.5 μm beads had an average velocity of 16.9 μm/min) (Fig. 7a, b and Supplementary Movies 4, 5). To further determine if MirA and N-WASP directly interact, a pulldown assay was employed using recombinant purified MirA and 6xHis-N-WASP. MirA was pulled down only when pre-incubated with N-WASP, suggesting the two proteins bind directly (Fig. 7c). To test if MirA activates N-WASP, thereby allowing N-WASP to bind the Arp2/3 complex to stimulate actin filament nucleation, we tested their activity in a pyrene actin polymerization assay. Purified MirA did not increase actin assembly when added to actin alone or with the purified Arp2/3 complex, supporting that MirA is neither itself an actin nucleator nor an Arp2/3-binding nucleation promoting factor (Fig. 7d and Supplementary Fig. 10a, b). Though, when combined with both purified N-WASP and the Arp2/3 complex, MirA accelerated actin assembly in concentration-dependent manner (>7.5-fold increase relative to N-WASP alone) (Fig. 7e and

Supplementary Fig. 10c) with 60 nM required for half-maximum activity (Fig. 7f). Thus, MirA promotes actin assembly by directly binding and activating N-WASP to stimulate Arp2/3-dependent actin polymerization.

## Discussion

Our work reveals that MirA is the *M. marinum* actin-based motility factor. We demonstrate that in response to infection, MirA is expressed, translocated, and anchored into the bacterial outer membrane through an amphipathic helix. MirA directly binds and activates N-WASP to stimulate actin filament nucleation through the Arp2/3 complex, which drives bacterial intracellular motility and cell-cell spread (see model in Fig. 7g).

MirA is a member of the PE_PGRS protein family, the largest group of proteins exported by virulent mycobacteria. Yet, the contribution of individual PE_PGRS proteins to infection is poorly understood, in part because these proteins lack sequence homology with other proteins in existing databases and have been recalcitrant to genetic or biochemical manipulation[31, 32]. PE_PGRS proteins are thought to primarily localize to the mycobacterial outer membrane, though how this localization is achieved has been an open question[31]. We demonstrate that MirA contains an amphipathic helix between the PE and PGRS

domains that is necessary and sufficient for localization to the bacterial surface. We also find that amphipathic helices are likely a common feature among the PE_PGRS protein family, possibly establishing a mechanism by which PE_PGRS proteins are targeted to the mycobacterial outer membrane. Further, we hypothesize that the bioinformatic prediction of amphipathic helices (Supplementary Data S1) may enable prediction as to which PE_PGRS proteins are localized to the mycobacterial surface or secreted freely into the host cytosol.

Surprisingly, we find that the amphipathic helix can also target MirA to the surface of lipid droplet organelles when ectopically expressed in eukaryotic cells. There, MirA stimulates actin polymerization that propels lipid droplets by actin-based motility, analogous to bacterial movement. This raises the possibility that MirA, which we observe in the host cytosol during infection, could be targeted to lipid droplets during infection. Recent work has shown that a host lipid droplet protein containing an amphipathic helix is transferred to the *M. marinum* surface during infection[33], suggesting that targeting of bacterial proteins to host lipid droplets may also occur. Further work will be necessary to determine if MirA and/or other PE_PGRS proteins are transferred to the host lipid droplet surface to impact infection, for example, by disrupting host immune signaling[34, 35] or enabling nutrient acquisition[36, 37].

PE_PGRS proteins are characterized by the PGRS domain, which is enriched for glycine residues, specifically GGX repeats. For instance, MirA has a ~529 amino acid PGRS domain composed of 51% glycine residues with 102 GGX sequences. Yet, it has been unclear how these glycine-rich sequences contribute to infection and whether they can target specific host proteins[38, 39]. A handful of the PE_PGRS proteins have been reported to contribute to functions related to host-pathogen interaction, such as suppressing autophagy[40, 41], inhibiting lysosomal fusion[42], and antagonizing signaling from Toll-like receptor proteins[25, 43–45]. Our discovery that MirA drives actin-based motility broadens the scope of host cellular processes targeted by PE_PGRS proteins.

We identify N-WASP as the specific host target of MirA using a non-biased approach that we confirmed biochemically. MirA's glycine-rich nature makes it unusual among eukaryotic WASP/N-WASP-binding partners. Curiously, IcsA, the *Shigella flexneri* actin-based motility effector that activates N-WASP, has a glycine-rich region important for stimulating actin-based motility[46]. The glycine-rich region in IcsA is substantially smaller (19 GGX repeats) in comparison to MirA (102 GGX repeats), but it may suggest convergent evolution between these two dissimilar bacterial pathogens. Importantly, elucidating how these glycine-rich regions promote N-WASP activation may enable the identification of previously unknown glycine-rich proteins that regulate the actin cytoskeleton.

We speculate that MirA regulates WASP/N-WASP through a previously undescribed mechanism. At resting state, N-WASP activity is autoinhibited by intramolecular interactions between the regulatory GTPase binding domain (GBD) and the WH2-central-acidic (WCA) domain[47–49]. Eukaryotic and microbial WASP/N-WASP-activating ligands bind to N-terminal regulatory domains that change the conformation and allows the WCA domain to bind the Arp2/3 complex and actin monomers. The four previously identified microbial factors that bind directly to N-WASP (*S. flexneri* IcsA, enterohemorrhagic *Escherichia coli* EspF_U, *Chlamydia trachomatis* TmeA, and the *Mycobacterium ulcerans* polyketide mycolactone) target proximal to the GBD of N-WASP[50–55]. Curiously, only the eight amino acid lysine-rich basic region of WASP/N-WASP, not the GBD, is necessary for *M. marinum* to stimulate actin-based motility[9]. This domain binds the phosphoinositol lipid PIP_2 to coordinate actin polymerization at cellular membranes[56, 57]. Why this domain is uniquely

important to *M. marinum* among pathogens that target WASP/N-WASP is unclear, but the discovery of MirA may provide a new model to understand how WASP family proteins can be regulated. Moreover, the MirA-N-WASP interaction represents a tractable model to understand how proteins in the PE_PGRS family are able to intercept host cell biological pathways to promote mycobacterial pathogenesis.

## Methods

**Bacterial strains, media, and plasmids.** *Escherichia coli* strains XL1-blue and BL21(DE3) were obtained from the UC Berkeley MacroLab and were used for cloning and protein purification, respectively. *E. coli* was cultured in lysogeny broth (LB), and antibiotics were used at concentrations of 100 μg/ml ampicillin, 50 μg/ml zeocin, or 50 μg/ml kanamycin when appropriate. Standard techniques were employed for cloning and other genetic manipulations. Full sequences of plasmids used in this study are provided in Supplementary Data 3a.

*M. marinum* strain M (NCBI:txid216594)[58] was cultured in Middlebrook 7H9 broth (Fluka, M0178) supplemented with 0.2% glycerol, 0.05% Tween 80, and 10% oleic acid-albumin-dextrose-catalase (OADC), or on 7H10 agar (Difco, 262710) plates supplemented with 0.2% glycerol and 10% OADC. When appropriate, antibiotics were added to the media at the following concentrations: 50 μg/ml hygromycin, 20 μg/ml kanamycin, and/or 30 μg/ml zeocin. *M. marinum* was grown at 33 °C and with shaking at ~100 rpm for liquid cultures. Electroporation to introduce DNA were conducted as previously described[59]. *M. marinum* strains used in this study are detailed in Supplementary Data 3b.

**Cell lines.** Mammalian cell lines U2OS (RRID:CVCL_0042), Raw 264.7 (RRID:CVCL_0493), HEK293 (RRID:CVCL_0045), and A549 (RRID:CVCL_0023) were obtained from and authenticated using short-tandem-repeat analysis by the University of California, Berkeley Cell Culture Facility (UCB-CCF). Cell lines were confirmed to be mycoplasma negative by DAPI staining and fluorescence microscopy screening at the UCB-CCF. Cells were grown at 37 °C in 5% CO_2 and maintained in DMEM (Gibco; 11965-092) containing 2–10% heat-inactivated and filtered fetal bovine serum (FBS) (Atlas Biologicals, FP-0500-A). Cells were not supplemented with oleate to induce production of lipid droplet organelles unless otherwise stated. Sf9 insect cells (RRID: CVCL_0549) were also obtained from the UCB-CCF and grown at 28 °C in in Grace's insect media (Gemini Bio-Products, 600-310) supplemented with 2% FBS (Gemini Bio-Products, 100-500) and penicillin/streptomycin. Bone marrow-derived macrophages from wild-type C57BL/6J mice were isolated and derived exactly as previously detailed[60]. Polyclonal U2OS cells stably expressing F-Tractin-mCherry[61] to mark F-actin or A549 cells expressing a farnesylated TagRFP to mark the plasma membrane were generated by lentiviral transduction as described previously[62].

**Transient transfections.** Cells were plated in 24-well plates 18–24 h prior at $1–1.5 \times 10^5$ cells/well such that cells were 60–80% confluent at the time of transfection. Plasmid DNA (500 ng) was diluted in Opti-MEM (Thermo, 31985062) and mixed with Lipofectamine 2000 (Thermo, 1668019) as per manufactures protocol. Cells were washed once with phosphate-buffered saline (PBS) (Gibco, 10010-023) and incubated in 300 μl Opti-MEM. DNA-Lipofectamine 2000 complexes were added to the cells for 2–4 h at 37 °C, then replaced with DMEM + 2% FBS. Samples were fixed or lysed at 24–48 h post-transfection. For transfections coupled to bacterial infection, cells were initially transfected for 4 h, washed, then infected with *M. marinum* and harvested 48 hpi.

For eukaryotic expression of *mirA*: wild-type (pBH261; *M. marinum* M (NCBI Reference Sequence: NC_010612.1) nts 4,401,156–4,403,237), ΔPE (pBH310; nts 4,401,447–4,403,237), and ΔAH (pBH351; nts 4,401,156–4,401,453 and 4,401,519–-4,403,237) variants of MirA were cloned into the NotI/BsrGI site of pBH257 with an added 5′ ATG and consensus Kozak sequence (GCCACC) and 3′ sequences encoding V5 and 6xHis epitope tags. To create the *mirA* amphipathic helix fused to tdTomato (pBH347), nts 4,401,447–4,401,549 from *M. marinum* genomic DNA was amplified by PCR and fused 5′ to the gene encoding *tdTomato* amplified from pBH251 as template. This fragment was then Gibson-cloned into the NotI/BsrGI site of pBH257 with an ATG start codon and Kozak sequence. In Fig. 5b, U2OS cells were transfected with pBH347, fixed at 24 h post-transfection (hpt), then stained with BODIPY 493/503. In Fig. 5c, U2OS cells were transfected with pBH347, washed, infected with BHm144 (Δ*mirA* + EBFP2), then fixed at 28 hpt/24 hpi. For Fig. 4a–e, U2OS cells were transfected with the parental vector (pBH257) or $P_{EF-1\alpha}$-*mirA*$^{261–2049}$-V5 (pBH310). For Fig. 4a, transfected cells were then stained for lipid droplets (BODIPY 493/503), MirA (anti-MirA), and F-actin (Phalloidin-iFluor 405). In Fig. 5d, e, U2OS cells were transfected with $P_{EF-1\alpha}$-*mirA*$^{\Delta297-363}$-V5 (pBH351) and stained for lipid droplets (BODIPY 493/503), MirA (anti-MirA), and either N-WASP (anti-N-WASP) or the Arp2/3 complex (anti-P34). For Supplementary Fig. 9c, MMAR_2645 (nts 3,221,630–3,223,507) was cloned into the NotI/BsrGI site of pBH257 with an ATG start codon, Kozak sequence, and a V5 epitope tag creating pBH443. Transfected U2OS cells were fixed at 24 h and stained for MMAR_2645-V5 (anti-V5) and lipid droplets (BODIPY 493/503).

**Bacterial infections of host cells**. The osteoclast cell line U2OS was used in some experiments for two reasons: they form flat, stationary, and confluent monolayers that allow for easy assessment of cell-to-cell spread; and they are non-hematopoietic cells that express N-WASP but not WASP. Host cells used for infection were seeded ~18–24 h prior to infection. U2OS cells were seeded into 96- and or 24-well plates at $4 \times 10^4$ or $2 \times 10^5$ cells per well, respectively. U2OS cells were incubated in DMEM without serum for ~60 min prior to infection to promote uptake of *M. marinum*. RAW 264.7 cells were seeded at $7.8 \times 10^6$ into T-75 flasks. BMDM were seeded into 96- or 24-well plates with $5 \times 10^4$ and $3 \times 10^5$ macrophages per well, respectively.

To prepare the bacteria for infection, *M. marinum* was recovered from −80 °C and resuspended into ~2–3 ml of 7H9-OADC at an $OD_{600}$ ~0.1–0.3. Cultures were grown for 18–24 h, then back-diluted into fresh 7H9-OADC to an $OD_{600}$ ~0.1 and grown for an additional ~18–24 h to an $OD_{600}$ ~0.8 to standardize growth phase prior to infection. Prior to infections, *M. marinum* was washed 3 times in 1 ml PBS, assessed for $OD_{600}$ to calculate MOI, then resuspended in DMEM + 10% serum. Bacteria were added to pre-seeded mammalian cells and immediately spinfected at $1200 \times g$ for 10 min at 22 °C. Infections were moved to a 33 °C incubator for 2 h before rinsing 3 times with PBS, then maintained in DMEM with 2–10% FBS with amikacin (40 μg/ml) to kill extracellular bacteria at 33 °C.

To measure infectious focus size, U2OS cells were plated onto 12 mm coverslips in 24-well plates. 18–24 h later, confluent monolayers were infected at an MOI of 1–20. Infection progressed for indicated time at 33 °C until fixation and staining. To quantify spread, individual foci were assessed for the number of infected host cells per focus where *M. marinum* was marked by expression of tdTomato from a multicopy plasmid (pBH251) and boundaries of host cells were determined by cortical actin staining via Alexa 488 phalloidin (A12379). For Fig. 1b, wild type (BHm102), *espG5::tn*-1 (BHm106), or *espG5::tn*-2 (BHm107) were used to infect U2OS cells and >12 foci were counted per each of three biological replicates. For Fig. 1c, d, wild type (BHm136), Δ*mirA* (BHm132), or Δ*mirA* + attB::$P_{mirA}$-*mirA* (BHm163) strains were used for infection of U2OS cells and >20 foci were counted per each of three biological replicates. For Supplementary Fig. 6d, wild type (BHm136) or Δ*mirA* + $P_{groEL}$-3581 (BHm152) were used to infect U2OS cells and >20 foci were counted per replicate over three biological replicates.

To measure the colocalization of *M. marinum* associated with N-WASP, MirA, and/or F-actin, U2OS or BMDM cells were plated on 12 mm coverslips and infected 18 h later at an MOI 1–50, depending on the duration of infection. Infected cells were incubated at 33 °C for the indicated time, fixed, then stained for N-WASP (anti-N-WASP), MirA (anti-MirA, this study), and/or F-actin (Alexa 488 phalloidin). Colocalization was manually scored using the ImageJ cell counter plugin with an actin structure at the bacterial >1 μm counted as an actin tail. For Fig. 2a, b, wild type (BHm103), Δ*mirA* (BHm144), and Δ*mirA* + $P_{mirA}$-*mirA* (BHm161) were used and 50–400 bacteria/replicate were calculated over three biological replicates. For Supplementary Fig. 3, wild type (BHm103) and Δ*mirA* (BHm144) were used to infect C57BL/6J BMDM cells, fixed at 48 hpi, then stained for F-actin using Alexa 568 phalloidin (Invitrogen; A12380). >300 bacteria were assessed for both strains for an actin tail for three biological replicates. For Supplementary Fig. 4, wild type (BHm103), Δ*mirA* (BHm144), Δ*mirA* + $P_{mirA}$-*mirA* (BHm170), or Δ*mirA* + $P_{groEL}$-3581 (BHm149) were used to infect U2OS cells. For Supplementary Fig. 4a, 50–250 bacteria/replicate were assessed for colocalization with both F-actin (Alexa 488 phalloidin) and N-WASP (anti-N-WASP) over three biological replicates. In Supplementary Fig. 4b, cells were fixed at 72 hpi and stained for F-actin (Alexa 488 phalloidin). To measure the bacterial cell length of *M. marinum* during infection in Supplementary Fig. 4c, 50–150 bacteria from each strain and time point were assessed for length using the freehand line tool in ImageJ[63] from three biological replicates.

To calculate bacterial growth during infection shown in Fig. 1e and Supplementary Fig. 2b, wild-type (BHm103) or Δ*mirA* (BHm144) bacteria were used to infect U2OS or BMDM host cells (in a 24-well plate format) at an MOI of 0.01 or 0.1, respectively. Prior to harvesting, infected cells were washed three times with PBS to remove extracellular bacteria. Host cells were then lysed in water for 10 min at 37 °C. From those lysates, serial dilutions were made in 7H9-OADC, plated onto 7H10-OADC + hygromycin agar plates, then enumerated 8–10 d later. Data represent three biological replicates. To calculate bacterial growth in media (Supplementary Fig. 2a), wild type (BHm101), Δ*mirA* (BHm129), and Δ*mirA* + $P_{mirA}$-*mirA* (BHm161) strains were cultured in 7H9-OADC broth at 33 °C/100 rpm and monitored by $A_{600}$ over two biological replicates.

For examining MirA colocalization with F-actin and N-WASP in Fig. 3b–d and Fig. 5f, Δ*mirA* (BHm144), Δ*mirA* + $P_{mirA}$-*mirA-flag* (BHm170), or Δ*mirA* + $P_{mirA}$-*mirA*$^{ΔAH}$-*flag* (BHm174) were examined during infection of U2OS cells. 50–300 bacteria were assessed for each strain at every time point over three biological replicates. For Fig. 5f, U2OS cells were infected with Δ*mirA* + $P_{mirA}$-*mirA*$^{ΔAH}$-*flag* (BHm174), fixed at 48 hpi, then stained for the FLAG epitope and F-actin (Alexa 488 phalloidin). In Fig. 3e, f, Δ*mirA* + $P_{mirA}$-*mirA-flag* (BHm170) was used to infect U2OS cells, and 100–150 bacteria were assessed at each time point for three biological replicates. For examining *in trans mirA* complementation of Δ*mirA* bacteria (Fig. 3g and Supplementary Fig. 7a, b), U2OS cells were transfected with a plasmid encoding *mirA-V5* (pBH261), *mirA*$^{ΔPE}$-*V5* (pBH310), *mirA*$^{ΔAH}$-*V5* (pBH351), or the parental vector control (pBH257). Transfections were washed and then immediately infected with *M. marinum* Δ*mirA* (BHm132), fixed at 48 hpi, and then stained for MirA-V5 (anti-V5) and F-actin using Phalloidin-iFluor

405 (Abcam; ab176752). For Supplementary Fig. 7a, 50–150 bacteria/replicate of each MirA variant were assessed for association with MirA or F-actin for three biological replicates. For Supplementary Fig. 5d, 100–200 bacteria per time point for both wild type (BHm102) and the Δ*mirA* mutant (BHm144) were assessed for significant colocalization with polyubiquitin during infection of BMDM host cells over three biological replicates.

To examine cytosolic assess for the Δ*mirA* mutant in Fig. 2c, d, intracellular *M. marinum* bacteria enveloped by a vacuole was examined similarly to previous studies[8, 64]. Briefly, Raw 264.7 host cells were infected with either wild type or Δ*mirA* strains expressing EBFP2. Two hours prior to indicated time point, host cells were incubated for 1 h with CM-DiI (Molecular Probes) at a final concentration of 2 μM. Infected host cells were then washed 3× with PBS to remove excess CM-DiI and incubated for another hour in fresh media to allow the dye to internalize from the plasma membrane to mycobacterial-containing vacuoles.

**Generating, screening, and mapping a *M. marinum* transposon insertion library**. A *M. marinum* transposon insertion library expressing mCherry was generated through transduction of the mycobacterium-specific phage phiMyco-MarT7 containing the mariner-like transposon Himar1 as previously described[65, 66]. This library had an estimated ~4.4 million insertions (~35-fold coverage of TA dinucleotide insertion sites in *M. marinum*). Individual colonies were isolated from 7H10-OADC plates containing kanamycin and hygromycin, grown in 80 μl of 7H9-OADC with containing kanamycin and hygromycin in 96-well plates, and used to infect near-confluent U2OS cells in a 96-well format. At 72 hpi, clones with defects in bacterial cell-to-cell spread were identified by fluorescence microscopy. Transposon insertions that caused defects in cell-to-cell spread due to a growth attenuation (e.g., insertions into genes of the ESX-1 secretory system) were not further examined.

Semi-random nested PCR was used to map the transposon insertion site of spread-defective mutants. To map the 5′ direction, oligos GCTTAGTACGTTAGCC ATGAGAGC and GGCCACGCGTCGACTAGTCANNNNNNNNNNNAGCTG were used in PCR with chromosomal DNA as the template. Subsequently, those PCR products were used as the template with oligos CACATTTCCCCGAAAAGTGCCAC and GG CCACGCGTCGACTAGTCA. To map the 3′ direction oligos TACCTGCCCATTCG ACCACCAAGC and GGCCACGCGTCGACTAGTCANNNNNNNNNNNAGCTG used in PCR with chromosomal DNA as the template. Subsequently, those PCR products were used as template with oligos CGCATCGCCTTCTATCGCCTTCTT and GG CCACGCGTCGACTAGTCA. ExoSAP-IT was then mixed with the second PCR and incubated for 30 min at 37 °C, then heat-inactivated. The amplicons were then sequenced with GGCCACGCGTCGACTAGTCA and genomic locations were determined using BLAST (GenBank/NCBI accession NZ_HG917972). A transposon insertion into *MMAR_3581* (*mirA*) was identified at nucleotide 4,403,232 (2046 in CDS) (strain BHm119). Two distinct transposon insertions into *MMAR_2680* (*eccA5*) were identified, at nucleotide 3,269,477 (1561 in CDS) (strain BHm106) or 3,269,674 (1758 in CDS) (strain BHm107).

**Construction of the *mirA* deletion and complementation strains**. The *M. marinum* Δ*mirA* strain was built using the *sacB/galK* counterselection method[67]. Briefly, ~1000 bp from both the 5′ or 3′ end of *mirA* was cloned into pBH92 resulting in pBH263 (sequence in Supplementary Data 3a). Wild-type *M. marinum* was electroporated with pBH263 and plated on 7H10-OADC with 50 μg/ml hygromycin, then screened for recombination by PCR. Strains with the initial recombination were grown to an $A_{600}$ of ~1.0 in 7H9-OADC and plated onto 7H10-OADC containing 5% sucrose and 0.2% 2-deoxy-galactose to select strains that had undergone the second recombination. Strains with a clean *mirA* deletion were identified by PCR and DNA sequencing.

The Δ*mirA* strain was complemented with MirA at two different expression levels. First, an overexpression strain was built by amplifying full-length *mirA* from *M. marinum* genomic DNA (nts 4,401,186–4,403,237) and cloned into pMV261[68] (pBH206) downstream of $P_{groEL}$ to create pBH417. Either $P_{MSP12}$-*ebfp2* (from pBH26) or $P_{MSP12}$-*tdtomato* (from pBH251) was then cloned into the PciI site to create pBH423 or pBH424, respectively. Second, a native MirA expression strain was built by inserting the *mirA* locus (nts 4,398,526–4,403,237) into the HindIII/KpnI site of the integrative vector pST-Ki[69] (pBH255) thereby creating pBH445. The *mirA* amphipathic helix deletion was built similarly, however it was amplified in two parts thereby omitting nts 4,401,492–4,401,548 creating pBH450. pBH445 or pBH450 were electroporated into Δ*mirA* expressing EBFP2 (BHm144) creating the BHm170 or BHm174 *M. marinum* strains.

**Immunoblotting**. For immunoblotting to evaluate protein expression, eukaryotic and *E. coli* cells were lysed in ice-cold RIPA buffer (Thermo, 89900) with protease inhibitors, protein concentrations were measured, then samples were mixed with 4× sample buffer and immediately cycled $3 \times 5$ min between a >90 °C water bath and ice. Lysates were electrophoresed by SDS-PAGE, then applied to PVDF membrane via semi-dry protein transfer. Membranes were then blocked in 5% milk, washed with PBS, then incubated with primary antibodies overnight at 4 °C. Primary antibodies were diluted in blocking buffer (2% BSA and 0.1% Tween-20 in PBS) as follows: GroEL (1:1000), GAPDH (1:10,000), N-WASP (1:1000), CDC-42 (1:750), WIPF2 (1:750), MirA (1:7500), His (1:1000), and Strep (1:1000) (sources of

antibodies are listed in Supplementary Data 3c). Washed membranes were then incubated in 1% milk with 1:5000 (or 80 ng/ml) secondary antibodies conjugated to HRP (Santa Cruz Biotechnology, sc-2005, sc-2357, sc-516102; Abcam, 6908; Pierce PA1-26848), then imaged with ECL western blotting detection reagents (GE Healthcare, RPN2106) using a Biomax Light Film (Carestream, 178-8207). To strip blots for reprobing, the PVDF membrane was submerged in 60 mM Tris, 2% SDS, and 0.8% ß-mercaptoethanol at 50 °C for 45 min, rinsed multiple times with distilled water then TBST, and re-blocked in 5% milk in TBST.

For immunoblotting of mycobacterial proteins, bacteria were isolated by centrifugation. Pellets were resuspended into PBS and heat-inactivated for 60 min at >90 °C. Bacteria were pelleted and resuspended in mycobacterial lysis buffer (15% sucrose, 50 mM Tris- pH 8.5, 50 mM EDTA), 1.0 mm silica beads were added, then cells were lysed using a Mini Beadbeater-8 Cell Disrupter (Biospec Products) at maximum power for $3 \times 40$ s. Protein concentrations were measured, then 4x sample buffer was added and cycled $3 \times 5$ min between a >90 °C water bath and ice. Lysates were then electrophoresed and immunoblotted as described above. For Fig. 3a, secreted MirA was assessed from Genapol X-080 supernatants from wild type (BHm103) and $\Delta mirA$ (BHm144) bacteria isolated from infected Raw 264.7 host cells. For Supplementary Fig. 6a, whole bacteria cell lysates of MirA$^{WT}$ (BHm144), MirA$^{+++}$ (BHm149), MirA$^{WT}$-Flag (BHm170), and MirA$^{\Delta AH}$-Flag (BHm174) were used to examine total MirA levels. For Supplementary Fig. 6b, plasmids encoding either empty (pBH257), MirA$^{WT}$ (pBH261), MirA$^{\Delta PE}$ (pBH310), or MirA$^{\Delta AH}$ (pBH351) were transfected into U2OS cells.

**Subcellular fractionation.** To determine MirA's subcellular localization (Supplementary Fig. 6c), wild type (BHm170) and $P_{mirA}$-$mirA^{\Delta AH}$ (BHm174) strains were used. Three x T75 flask of Raw 264.7 cells were infected at an MOI of 2.5 for 3 h. After 48 h, infections were washed with PBS, then scraped and centrifuged at $3200 \times g$, 5 min, 4 °C. Cell pellet was washed once then resuspended in cold 250 µl of K36 buffer (50 mM K$_2$HPO$_4$, 100 mM KCl, 15 mM NaCl, pH 7) with protease inhibitors and Dounce homogenized with ~60 strokes. Lysates were then initially spun at $200 \times g$ for 5 min at 4 °C to pellet host cell debris. Supernatants were collected and spun at $7000 \times g$ for 1 min at 4 °C to pellet bacteria. The supernatant was collected and used as the "host cytosolic" fraction. The pellet was subsequently washed 3× in 1 ml PBS, then resuspended and incubated in 200 µl of K36 buffer with protease inhibitors and 0.5% Genapol X-080 (Sigma; 48750) for 45 min at 37 °C. Bacteria were pelleted $7000 \times g$ for 1 min at 4 °C and the supernatant was collected and used as the "genapol-extracted" fraction. The bacterial pellet was then washed 3 times in 1 ml PBS. The bacterial pellet was then lysed as described above.

**Fluorescence microscopy.** Eukaryotic cells affixed to 12 mm glass coverslips in 24-well plates were fixed with 4% paraformaldehyde in PBS for 10 min at 22 °C. Cells were then permeabilized with 0.5% Triton X-100 for 5 min, washed, then blocked with blocking buffer (2% BSA and 0.1% Tween-20 in PBS) for 20 min at 22 °C. Primary antibodies, which are listed in Supplementary Data 3c, were incubated with samples overnight at 4 °C in a humidified chamber. Primary antibodies were diluted in blocking buffer as follows: anti-N-WASP (1:750), anti-CDC-42 (1:1000), anti-P34 (Arp2/3 subunit) (1:500), anti-MirA (1:1000), anti-polyubiquitin (1:250), anti-V5 (1:1000), anti-His (1:1000), and anti-FLAG (1:1000). Secondary antibodies, conjugated to Alexa fluorophores (Invitrogen; A21131, A11008, A11073, A11075, A11036, A11004) were used at 1:1000 in blocking buffer. F-actin was stained using Alexa 568 phalloidin or Alexa 488 phalloidin at 1:400, or Phalloidin-iFluor 405 at 1:250, in blocking buffer. For experiments staining lipid droplets, cells were permeabilized, blocked, and maintained in 2% BSA with 0.5% saponin in PBS. BODIPY 493/503 (Thermo, D3922) was added at a concentration of 1 µM, and coverslips were mounted in 20 mM Tris, 0.5% N-propyl gallate, 85% glycerol, pH 8. Images were captured on a Nikon Ti Eclipse microscope with a Yokogawa CSU-XI spinning disc confocal, ×60 and ×100 (1.4 NA) Plan Apo objectives, a Clara Interline CCD Camera and MetaMorph software. Images were processed and quantified using ImageJ and assembled in Adobe Illustrator. For Supplementary Movies 1, 2, and 5, manual tracking using ImageJ was used to trace the trajectories of moving objects.

**Movement parameters of *M. marinum* and lipid droplets.** The velocities of *M. marinum* and lipid droplets were measured using U2OS cells stably expressing F-tractin-mCherry. Cells were seeded at $6 \times 10^5$ in 20 mm MatTek glass-bottom dishes, incubated for 18–24 h then either infected with *M. marinum*-expressing GFP or transfected with a plasmid encoding with $mirA^{\Delta PE}$-V5 (pBH310). Bacteria were evaluated 48–72 hpi, while lipid droplets were imaged 24–48 hpt. Live-cell imaging was conducted in a 33 °C environmental chamber with a Nikon Ti Eclipse with ×100 objective at 5 s intervals for 5–10 min. Only lipid droplets with a diameter similar to the width of mycobacteria ($0.35 > x > 0.75$ µm) were evaluated. To calculate actin-tail mediated velocities, the manual tracking plugin in ImageJ was used to track movement >10 consecutive frames (52 tracks were collected for *M. marinum* and 88 tracks for lipid droplets over three experiments). The efficiency of movement was measured by calculating final displacement divided by the total distance traveled over a 100 s time interval ($n = 78$).

**MirA's effect on host lipid droplet size, biogenesis, and cell-to-cell spread.** To assess the effect on lipid droplet size and number in response to ectopic expression of MirA, A549 cells were transfected as described above with either a plasmid encoding $mirA$-V5 (pBH261), $mirA^{\Delta PE}$-V5, $mirA^{\Delta AH}$-V5 (pBH351), or the parental control plasmid (pBH257). Transfected cells were washed after 4 h and maintained in DMEM + 2% FBS without oleate. At 24 hpt cells were fixed with 4% PFA, permeabilized, blocked, and maintained in 2% BSA with 0.5% saponin in PBS as described above. Cells were incubated with 1:1000 anti-V5 overnight. Secondary 1:1000 goat anti-mouse Alexa Fluor 568 and 1 µM BODIPY 493/503 (Thermo, D3922) were added in in 2% BSA with 0.5% saponin in PBS. Coverslips were mounted in 20 mM Tris, 0.5% N-propyl gallate, 85% glycerol, pH 8. Z-sections at 0.4 µm were captured through the entire cell and then Z-projected at max intensity on ImageJ.

For monitoring lipid droplet cell-to-cell spread, A549 cells stably expressing farnesylated-RFP to mark the plasma membrane (donor cells) were plated in 24-well plates. 18–24 h after plating, donor cells were transfected with pBH310 ($P_{EF-1a}$-$mirA^{\Delta PE}$). At 4 hpt, donor cells washed 3× PBS, lifted with 0.25% trypsin. Donor cells were then mixed with unmarked, untransfected A549 (recipient cells) at a ratio of 1:10 and plated on glass coverslips in a 24-well plate in DMEM + 2% FBS and 250 µM of oleic acid. Plates were placed in a humidified secondary container in the incubator to promote even cell distribution. Cells were incubated for 36 h at 37 °C prior to fixation and staining.

**Affinity purification and mass spectrometry.** Affinity purification and mass spectrometry were performed similarly to previously described[70]. $mirA$ without the PE domain (nts 4,401,447–4,403,237) was amplified by PCR from *M. marinum* genomic DNA with a C-terminal One-STrEP tag and cloned into the NotI/BsrGI site of pBH257 resulting in pBH342. In addition, the amphipathic helix mutant was built similarly except $mirA^{\Delta AH}$ was generated from PCR with pBH351 as template to create pBH364. pBH342 or pBH364 and the parental vector (pBH257) were separately transfected into HEK293 cells in $4 \times 8$ cm plates using calcium phosphate-mediated transfection. At 42 hpt, cells were scraped, pelleted for 5 min at $1200 \times g$ and 4 °C, resuspended in cold lysis buffer (50 mM Tris–HCl, 150 mM NaCl, 1 mM EDTA, 0.5 % IGEPAL CA-630, protease inhibitors, pH 7.4), then lysed using a Dounce homogenizer. Lysates were clarified by centrifuging for 20 min at $2800 \times g$ and 4 °C. Supernatants were initially incubated with 120 µl preclearing beads (Sepharose FF) for 2 h with rotation at 4 °C. The precleared lysates were immobilized with Strep-Tactin resin (15 µl/8 cm plate) for 4 °C for 18 h with rotation. The resin was washed 5× lysis buffer, and twice in lysis buffer without detergent before elution in that buffer containing 2.5 mM desthiobiotin (40 µl/8 cm plate). The eluate was analyzed by SDS-PAGE and either Coomassie or silver staining prior preceding with trypsin digestion. To use N-WASP as bait, full-length N-WASP was amplified from pBH5 adding 5′ sequences encoding a twin strep tag using an oligonucleotide primer. Strep-N-WASP was inserted into the XbaI site (under $P_{CMV}$) of pBH310 that co-expressed $mirA^{\Delta PE}$-V5 creating pBH359. Experimental conditions were as described above, then eluates were probed for N-WASP and MirA via immunoblot. For Fig. 6a, b, HEK293 cells were transfected with a plasmid encoding the empty parental vector (pBH257) or $mirA^{\Delta PE}$-$strep$-$his$ (pBH342). For Fig. 6c, HEK293 cells were transfected with a plasmid encoding either the empty parental vector (pBH257), $mirA^{\Delta PE}$-$strep$-$his$ (pBH342), or $mirA^{\Delta PE,\Delta AH}$-$strep$-$his$ (pBH364). For Fig. 6d, HEK293 cells were transfected with a plasmid encoding either $mirA^{\Delta PE}$-V5 (pBH310) or $mirA^{\Delta PE}$-V5 + $strep$-$n$-$wasp$ (pBH359).

For subsequent mass spectrometry, proteins eluates were digested with trypsin for LC-MS/MS analysis. Samples were denatured and reduced in 2 M urea, 10 mM ammonium bicarbonate, 2 mM DTT at 60 °C for 25 min then immediately alkylated with 2 mM iodoacetamide for 30 min in the dark at 22 °C. Trypsin was added at a 1:100 trypsin: substrate ratio and digested overnight at 37 °C. Samples were desalted using C18 ZipTips (Millipore) then speed-vacuumed to desiccation and finally resuspended in 0.1% formic acid.

Digested peptide mixtures were analyzed by LC-MS/MS on a Synapt G2-Si mass spectrometer, which was equipped with a nano electrospray ionization source (Waters). The mass spectrometer was connected in line with an Acquity M-class ultra-performance liquid chromatography system equipped with trapping (Symmetry C18; inner diameter, 180 µm; length, 20 mm and particle size, 5 µm) and analytical (HSS T3; inner diameter, 75 µm; length, 250 mm; particle size, 1.8 µm) columns (Waters). Data-independent, ion mobility-enabled, high-definition mass spectra and tandem mass spectra were acquired in the positive ion mode. Data acquisition was controlled using MassLynx software (version 4.1), and tryptic peptide identification and relative quantification using a label-free approach were performed using Progenesis QI for Proteomics software (version 4.0, Waters). The raw data were matched to protein sequences by the Protein Prospector algorithm[71]. Data were searched against a database containing SWISS-PROT Human protein sequences and concatenated to a decoy database where each sequence was randomized to estimate the false positive rate (accessed 28 August 2018). Protein species with a normalized spectral abundance factor of over 0.00025 were then scored with Mass spectrometry Interaction STatistics (MiST) algorithm, using the MiST reproducibility (0.45), specificity (0.50), and abundance (0.05) weights[28]. MirA-prey pairs with a MiST score ≥0.75 were deemed confident interactions and were combined with human protein interactions from the

STRING databases[72]. The resulting network diagram from two biological replicates was plotted using Adobe Illustrator in Fig. 6a (for further detail see Supplementary Data 2).

**Protein purification.** To generate MBP-fused MirA for initial affinity purification, *mirA* (pBH312; nts 4,401,156–4,403,237) or *mirA$^{\Delta 1-297}$* (pBH324; nts 4,401,447–4,403,237) were PCR-amplified from *M. marinum* genomic DNA and cloned in-frame into the SspI site of 6XHis-MBP-TEV (AddGene: 29656). Plasmids were freshly transformed into the *E. coli* BL21(DE3) background and MirA expression was induced at 37 °C with 0.5 mM IPTG for 3 h. His-MBP-MirA were first isolated using amylose bead affinity chromatography (NEB) in conditions specified by the manufacturer. His-MBP was cleaved from MirA overnight at 4 °C in the presence of the His-TEV at a molar concentration of 1:100. MirA was purified away from His-MBP by gel-filtration chromatography using a Superdex 200 column in 20 mM Tris–HCl (pH 7.5), 300 mM NaCl, 10% glycerol, 0.5 mM DTT, then snap-frozen in liquid nitrogen.

His-N-WASP was expressed and purified using the baculovirus expression system. His-N-WASP was PCR-amplified from pBH5 using primers encoding a 6×-his and cloned into a bacmid resulting in pBH354. Recombinant baculovirus was generated in Sf9 cells using the Bac-to-Bac system according to the manufacturer's instructions (Invitrogen, Carlsbad, CA). Recombinant proteins were then expressed by infecting Hi5 cells for 96 h at 28 °C. Hi5 cells were pelleted, resuspended in lysis buffer (20 mM NaH$_2$PO$_4$, 400 mM NaCl, 20 mM imidazole, 10 mM BME, 1% IGEPAL, 25 units/l benzoate nuclease, protease inhibitors, pH 7.8), incubated for 15 min on ice, then sonicated at 10% power 4 × 15 s. Lysates were spun at 20 K × *g* at 4 °C for 45 min. His-N-WASP was first isolated using Ni-NTA beads (Qiagen, Valencia, CA) using a buffer with an imidazole gradient (20 mM Bis-Tris, 500 mM NaCl, 1 mM TCEP, pH 6.5, imidazole 20 mM to 500 mM). Fractions were then concentrated and flowed through a 5 ml SP SH cation exchange column (GE Healthcare) to a cation exchange column (20 mM Bis-Tris, 100 to 1000 mM KCl, 2 mM MgCl$_2$, 0.5 mM EGTA, 0.5 mM EDTA, pH 6.5). Fractions were then subjected to gel filtration using a Superdex 200 (GE Healthcare) column into Control Buffer (20 mM Bis-Tris, 400 mM KCl, 2 mM MgCl$_2$, 0.5 mM EGTA, 0.5 mM EDTA, 0.5 mM DTT, 10% glycerol, pH 6.5). Fractions were concentrated and snap-frozen in liquid nitrogen.

**MirA antibody generation.** Polyclonal antibodies were generated against MirA in a rabbit host. The native full-length MirA protein was purified as described above and sent to Pocono Rabbit Farm and Laboratory (Canadensis, PA) where a 91-day custom antibody protocol was performed. To affinity-purify MirA specific antibodies from serum, purified MirA$^{\Delta PE}$ (described above) was dialyzed into ligand coupling buffer (200 mM NaHCO$_3$, 500 mM NaCl, pH 8.3) and then coupled onto NHS-ester Sepharose 4 Fast Flow resin (Cytiva; 17-0906-01). The resin was incubated with serum overnight at 4 °C with rotation, washed, and eluted with low pH buffer (100 mM Glycine, pH 2.5). Eluted fractions were neutralized with 1 M Tris pH 8.8 and dialyzed into PBS overnight at 4 °C. Affinity-purified antibodies were then concentrated using a 10 K MWCO Amicon filter (MilliporeSigma; UFC801096) and stored at −80 °C.

**Bulk actin assembly assays.** Rabbit skeletal muscle actin was purchased (unlabeled and pyrene-labeled actin; Cytoskeleton Inc.). All actin was maintained in G buffer (5 mM Tris, pH 8.0, 0.2 mM CaCl$_2$, 0.2 mM ATP, 0.5 mM DTT) prior to polymerization assays. Pyrene actin polymerization reactions were started by combining monomeric actin in G buffer (at a final concentration of 1 μM actin (10% pyrene-labeled)) with a mix of MirA, 10X initiation buffer (10 mM MgCl$_2$, 10 mM EGTA, 5 mM ATP, 500 mM KCl) and MirA buffer (see purification details). Fluorescence was detected at 20 s intervals for 1 h on a Tecan Infinite F200 Pro plate reader using a 365 nm excitation filter (10 nm bandpass), a 405 nm emission filter (20 nm bandpass) and Magellan software (v. 7). The reported reaction curves were normalized to the minimum fluorescence value of each reaction. The time for each reaction to reach the half-maximum fluorescence was determined following normalization to both the minimum and maximum fluorescence values, averaged from a minimum of three experiments as previously described[73]. The means are reported with their standard deviations. Data were analyzed in Excel, and the final graphs were constructed in PRISM v. 8 and v. 9 (GraphPad Software, Inc). For Fig. 7d and Supplementary Fig. 10a, 1 μM actin (10% pyrene-labeled) was mixed with initiation buffer alone or with 200 nM MirA, 200 nM MirA + 50 nM Arp2/3, 200 nM MirA + 200 nM N-WASP. Data in Supplementary Fig. 10b are mean ± SD from 3 technical replicates. For Fig. 4e, f, 1 μM actin (10% pyrene-labeled) was mixed with either 50 nM Arp2/3, 50 nM Arp2/3, and 200 nM N-WASP, or 50 nM Arp2/3, 200 nM N-WASP, and increasing concentrations of MirA. Data in Fig. 4f are mean ± SD from 3 technical replicates.

**Pulldown assay.** For Fig. 7c, purified native MirA (prey) and 6xHis-N-WASP (bait) were purified as described above. 50 μg of His-N-WASP was incubated alone or with 50 μg of MirA in 400 μL of for 2 h at 4 °C under agitation. His-N-WASP ± MirA was loaded onto a column with 80 μl of previously equilibrated Ni-NTA resin (Qiagen) and incubated for 1 h at 4 °C with agitation. Column was centrifuged for 1 min at 1200 × *g*. Beads were washed thrice with 20 mM Tris–HCl,

pH 7.5, 250 mM NaCl, 20 mM imidazole, then eluted with 100 μl of the same buffer except with 500 mM imidazole. Eluates were evaluated by immunoblot because MirA and N-WASP could not be resolved from each other as they migrate at a similar molecular weight.

**Actin polymerization on polystyrene beads in *Xenopus laevis* extract.** Polystyrene microspheres (Fluoresbrite™ Carboxylate YG, 0.5 μm and 1 μm (Polysciences) were incubated on ice with 5 μM MirA for >1 h before adding BSA to a concentration of 5 mg/ml and incubating for 15 min. Beads were washed in CSF-XB (10 mM HEPES at pH 7.7, 2 mM MgCl$_2$, 0.1 mM CaCl$_2$, 100 mM KCl, 5 mM EGTA and 50 mM sucrose) and kept at 4 °C. *Xenopus laevis* egg extract was provided by the laboratory of Dr. R. Heald (University of California, Berkeley). To 8 μl of extract, 1 μl of actin (3 μM, 20% of which was rhodamine-labeled) and 1 μl of BSA- or MirA-coated beads were added, and 1–2 μl of this dispersion was placed between a slide and coverslip and observed immediately by epifluorescence microscopy. ImageJ was used to adjust brightness and contrast and export QuickTime movies. To calculate bead velocities, the manual tracking plugin in ImageJ was used to track movement 10 consecutive frames (every 2.5 s for the 0.5 μm beads and 5 s for the 1 μm beads). >40 counts over three replicates were assessed and shown in Fig. 7b.

**Transmission electron microscopy.** For examining *M. marinum* escape from the mycobacterial-containing vacuole by electron microscopy (Supplementary Fig. 5a, b), *M. marinum* wild-type (BHm102) and *mirA::tn* (BHm119) strains expressing mCherry were used to infect U2OS host cells (MOI = 10) in 6-well plates. At 60 hpi, cells were scraped, pelleted at for 5 min at 1200 × *g* at 22 °C, then gently resuspended in fixative (1.5% paraformaldehyde, 2.0% glutaraldehyde, and 0.03% CaCl$_2$ in 0.05 M cacodylate buffer, pH 7.2). After 45 min at 22 °C, the samples were embedded in 2.0% glutaraldehyde and 0.1 M cacodylate buffer, and 2% low-melting agarose and incubated overnight at 4 °C. The samples were post-fixed the following day in 1% osmium tetraoxide and 1.6% potassium ferricyanide, dehydrated in a graded series of ethanol concentrations, and embedded in EPON 812 resin prepared as follows: 11.75 g eponate 12, 6.25 g dodecenyl succinic anhydride, and 7 g NADIC methyl anhydride were mixed before the addition of 0.375 ml of the embedding accelerator benzyldimethylamine. Next, the samples were stained with 2% uranyl acetate and lead citrate. Images were obtained on a FEI Tecnai 12 transmission electron microscope and processed on ImageJ. To quantify *M. marinum* localized within a vacuole, two biological replicates of wild type (BHm102) and *mirA::tn* (BHm119) were assessed with >65 bacteria per replicate.

**Quantification and statistical analysis.** Statistical parameters and significance are reported in the figures and the figure legends. Data are determined to be statistically significant when $p < 0.05$ where indicated. As such, asterisks denote statistical significance as: $*p < 0.05$; $**p < 0.01$; $***p < 0.001$; $****p < 0.0001$, compared to indicated controls. All other graphical representations are described in the figure legends. Statistical analysis was performed in GraphPad PRISM v. 8 and v. 9.

**Reporting summary.** Further information on research design is available in the Nature Research Reporting Summary linked to this article.

## Data availability

Genomic locations of transposon insertion were determined using BLAST against GenBank/NCBI accession NZ_HG917972. Data acquired by mass spectrometry (Fig. 6a) were searched against a database containing SWISS-PROT Human protein sequences (accessed 28 August 2018) and examined by the STRING database (accessed 10 May 2020). All other data that support the findings of this study and a detailed description of the methods used are available in the manuscript, Supplementary Information, or source data. Source data are provided with this paper.

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

## Acknowledgements

We thank the laboratories of Sarah Stanley (UC Berkeley), David Tobin (Duke Univ.), James Olzmann (UC Berkeley), Jeffery Cox (UC Berkeley), and Christina Stallings (Washington Univ.) for providing us with reagents, equipment, and/or expertise. We thank Erin Benanti for purification of the Arp2/3 complex and generating the F-tractin-mCherry U2OS cell line, Taro Ohkawa for expertise in insect protein purification, Rebecca Lamason for generating the RFP-CAAX A549 cell line, Thomas Burke and Patrik Engström for generating BMDM cells, Daniel Portnoy, Neil Fischer, and Mary Anne Hill for their critical reading of this manuscript, Bradley Kraushaar and Erin Brosnahan and for their assistance in Adobe Illustrator, and current and former Welch lab members for their invaluable contributions. We also thank UC Berkeley core facilities, including Ann Fisher and Alison Killilea (Cell Culture Facility), Lori Kohlstaedt (Vincent J. Proteomics/Mass Spectrometry Laboratory), and Mary West (CTAF/HTSF). This work was supported by grant R35 GM127108 from the NIH/NIGMS to M.D.W. and Jane Coffin Child Fund Fellowship to N.S.H.

## Author contributions

N.S.H. and M.D.W.: Conceptualization, methodology, funding acquisition, project administration, and writing—review & editing. N.S.H.: Investigation, visualization, and writing—original draft. Supervision: M.D.W.

## Competing interests

The authors declare no competing interests.
