## [Peer Review File · Nature Communications]

Reviewers' Comments:

Reviewer #1:

Remarks to the Author:

The manuscript by Norbert S. Hill & Matthew D. Welch titled "A mycobacterial glycine-rich protein governs actin-based motility" is an original work that points the identification and characterization of MirA an *M. marinum* actin-based motility factor, a member of the PE_PGRS family of glycine-rich of ESX-5-secreted proteins. The novelty of this work relies on:

(1) The fact that MirA while binding directly to WASP/NWASP to recruit the actin nucleating factor ARP2/3 similar to that described in *Shigella flexneri* IcsA, enterohemorrhagic *Escherichia coli* EspFU, and *Chlamydia trachomatis* TmeA it does not target the GTPase binding domain indicating a distinct role of actin nucleator or actin promotor. Instead MirA targets a small positively charged basic region only known to interact with a phosphoinositol lipid (PIP2) with potential to represent a new class of microbial and host actin regulator.

(2) MirA uses an amphipathic helix to anchor into the mycobacterial outer membrane and, also the surface of host lipid droplet organelles indicating a relevant virulent factor in manipulating the host lipid droplets and potential associated vesicles intracellular trafficking representing a model to understand how the PE_PGRS proteins contribute to mycobacterial pathogenesis.

While the gene product MMAR_3581 looks to be encoded only by *M. marinum* among mycobacterial species with a truncated form encoded by the closely related pathogen *Mycobacterium ulcerans* it cannot be excluded the possibility to similar genes operating in *M.tuberculosis*.

As the authors stated the PE_PGRS family of proteins at the level of individual contribution to infection is poorly understood due to lack of sequence homology.

According to KEGG orthology database there's a good match between the MMAR-3581 and the *Mtb* ERdman strain mtn: ERDMAN_3853 ortholog while no actin based motility was demonstrated so far on virulent *Mtb* escaping to the cytosol in any model used of host cells (https://www.kegg.jp/ssdb-bin/ssdb_best?org_gene=mmi:MMAR_3581; GenBank: BAL67626.1). It would be interesting to assess the effects of the ortholog gene product of the Erdman strain of *Mtb* in a context where actin tail formation is facilitated the ortholog protein will also display a similar phenotype.

The impact of the present work is that this family of proteins are related with important virulence factors of pathogenic mycobacteria: a similar system in *Mtb* will help bacteria spread to new non infected cells and in addition the ability to induce actin tails in host lipid droplets may be involved in host lipid droplets trafficking and signaling for host manipulation while promoting bacterial survival.

The work here shown is original and sound. The work is well supported with methods and full demonstration of the enrolment of MirA in recruitment of WASP/NWASP and interacting proteins either to the surface of the bacteria either to host lipid membrane droplets. The experiments are robust and well controlled with lots of supporting experiments

Curiously the authors demonstrate that MirA targets a charged region of the phospholipid layer that interacts with PIP2. While it is described the interaction of WASP with PIP2 (PMID: 23445807) there are other ABP described to serve as intermediated in the recruitment of actin via PIP2 to host vesicle membranes to induce the formation of actin filaments. The ERM family of proteins were previously shown relevant to induce the formation of actin tracks from phagosomes containing latex beads (PMID: 10637224) and similar system was suggested to occurs in phagosomes containing non-pathogenic and pathogenic mycobacteria contributing to the pathogenesis of *Mtb* and *M. avium* (PMID: 12942085). While this event is related with the phagosomal context and membranes maybe this should be explored in the discussion for a possible relationship between the actin associated membranes and PIP2, ERM and a possible new role of MirA in promoting actin filamentation .

The work is robust and suited for publication taken in consideration a few points outlined to introduce in the discussion.

Reviewer #2:

Remarks to the Author:

In this study, Hill and Welch identify a *M. marinum* protein (MirA) that promotes actin polymerization in infected cells. Starting from a *M. marinum* mutant library the authors identify the gene MMAR_3581 as a gene required for proper *M. marinum* spread in infected cells. Designing very elegant experiments the authors demonstrate that MirA is necessary to activate actin polymerization by interacting with the N-WASP and the Arp2/3 complex. The results of this study shed new light on the mechanisms used by *M. marinum* to spread in infected cells and most importantly implicate a member of a family of important mycobacterial proteins (PE_PGRS) in the pathogenesis of mycobacterial infections.

Major comments:

The authors conclude that the amphipathic linker AH region is necessary and sufficient for localization of MirA on the bacterial surface. However, there are few issues that in my opinion shall be clarified before reaching this conclusion.

This conclusion implies that the PE domain is cleaved off from the full length MirA protein, with cleavage occurring immediately after the PE domain (somewhere between position 88-101). However, the data provided to support cleavage at this site are weak, since the WB in extended Fig 3A indicate that most of the protein is intact and not cleaved. These results are important considering previous studies indicating that, in *M. marinum* and in *Mtb* (Daleke 2011, Burgraaf 2020, De Maio 2021), the cleavage site of the PE domain in PE_PGRS proteins is downstream of what indicated in this study.

It is also important to clarify whether cleavage of the PE domain is necessary to allow the MirA amphipathic AH domain proper localization on mycobacterial membrane and lipid droplets. A better characterization of this aspect is needed to support the hypothesis: comparison between the MirA expressed in *M. marinum* and that ectopically expressed in transfected cells may be useful to clarify this issue. Moreover, subcellular fractionation may provide insights on the cleavage of the PE domain and stability of the MirDeltaAH mutant (see comment below).

From what indicated in the methods, the MirDeltaAH functional deletion mutant was generated by omitting the sequence coding for the linker region (residues 102-119). It would be interesting to know whether the lack of this linker domain in the mutant affect protein stability and secretion through the mycobacterial membrane. This is an important domain and direct fusion between the PE domain and the region upstream the AH linker may affect the protein functional properties.

Minor comments:

Page 6, line 6: Rather than using the term "PE family domain" I suggest "PE domain".

Page 7, line 18: PE domain rather than PE family; moreover, PGRS domain rather than the plural domains.

Page 8, line 4: probably endogenously rather than endogenous.

Page 10, line 16: "... MirA is the *M. marinum* actin-based motility factor". It may be more appropriate to state that "... MirA is a *M. marinum* actin-based motility factor"

Page 37, line 20: "... PE super domain..". PE domain.

Reviewer #3:

Remarks to the Author:

A. General appreciation

Hill & Welch make substantial efforts to investigate the molecular mechanisms underlying actin polymerization at the surface of *Mycobacterium marinum* to promote intracellular mobility and dissemination to neighboring cells during infection. To this end, *M. marinum* recruits the host proteins WASP and/or N-WASP to stimulate actin nucleation by the Arp2/3 complex.

The main findings are that (i) *M. marinum* secretes the factor MirA, a member of the PE_PGRS family of Esx-5 secreted proteins, to recruit and activate N-WASP and that (ii) MirA is anchored to both, the surface of *M. marinum* and host lipid droplets (LDs) via its amphipathic helix to promote bacterial but also LD actin-based motility. The findings are new and will be of significance for the

field since only few PE_PGRS proteins have been described to be involved in host-pathogen interactions, however the molecular mechanism remain, in contrast to the here characterized MirA, elusive. The study meets high quality standards. The microscopy images and videos, in particular, are of exceptional quality. I have mostly minor comments.

B. Major comments

1. To rule out that the defect in actin-based motility of the delta_mirA mutant was not due to a problem of cytosolic access, the authors carried out an experiment that is based on electron microscopy (Fig. 1g, h). Because quantifying dynamic processes like phagosomal escape using 2D electron microscopy is challenging and also the escape kinetics of *M. marinum* in U2OS cells are not known the authors should analyze phagosome escape also with another method (for example co-localization with ubiquitin or xenophagy markers) and include more time points. Only one late time point was chosen (60 hpi), which makes it incomparable to other experiments (examples: Fig. 1b, 1e -> 48 hpi)
2. Page 4, lines 12,13: bacteria were assessed for association with actin clouds, actin tails, and N-WASP over the course of infection. Please explain how actin clouds/actin tails/N-WASP were defined and quantitated (manually or automatically?)
3. To be included in the discussion: MirA appears to be mainly enriched at one bacterial pole but completely surrounds host LDs. Please include a possible explanation (lipid composition, membrane curvature...) also in the context that actin polymerization takes place at one pole of *M. marinum* and at one concentrated spot on the LD surface.
4. Page 7, line 17: Proline is thought to destabilize amphipathic helices. Fig. 3f. shows the putative amphipathic helix of MirA including a proline. Please include in the results section or the discussion that this proline might disrupt the helix.
5. Perilipins are LD proteins that have mainly protective function (to shield the hydrophobic LD core from cytosolic lipases) and are targeted to the phospholipid monolayer of LDs via an amphipathic helix. The role of MirA on the LD surface remains speculative; therefore, it would be interesting to know if the number of host LDs is changed upon MirA expression. Does MirA expression induce the formation of LDs?
6. Concerning the actin polymerization assays: by comparing both figures (Fig. 4f, 4g) and by following the text it is difficult to understand the differences between both experiments. The authors should change the figure legends to make it more implicit. Only one representative experiment is shown (for both experiments in Fig. 4f, 4g), therefore the authors should include a second representative in the supplementary data.
7. Discussion: Page 12, line 10: In the manuscript it was not formally shown that the MirA PGRS domain directly binds and activates N-WASP. Therefore, the authors should formulate the sentence more cautiously.
8. Page 12, lines 19-21: "However, the GBD of WASP/N-WASP is dispensable for *M. marinum* 20 actin-based motility, and instead MirA targets a small positively charged basic region only known to interact with a phosphoinositol lipid (PIP2)" As this was not directly demonstrated here, the authors should formulate this more cautiously.

C. Minor comments

9. U2OS are not commonly used for infections. Please explain why U2OS cells were used in this study.
10. Wild-type or wild type, please decide for one and check throughout
11. To facilitate reading. In panels with microscopy images, please indicate the times post infection.
12. It is not clear if LDs were induced prior to the LD experiments in various cells types (A549, U2OS, HEK293), please mention briefly in M&M or in the figure legends.
13. Page 1, line 21: MirA is an abbreviation, please write the full name once in the abstract
14. Page 4, line 10: remove comma after e.g.
15. Page 5, line 8: 2x that, delete one
16. Page 5, line 19: apparently here "Percent of MirA-positive bacteria colocalizing with N-WASP or an actin tail "was measured. Please change the text of the manuscript accordingly and remove "that colocalized with MirA".
17. Page 6, line 16: Fig. 2h, white arrow (not 2g)
18. Page 8, L. 13: MirA uses an amphipathic alpha helix to insert in the phospholipid monolayer of both host LDs and the mycobacterial outer membrane. The authors should be cautious as the

mycobacterial outer membrane composition is quite complex. I would recommend to write that it might resemble a monolayer.

19. Page 9, lines 8 and 11: (This approach also identified....) Remove parenthesis.

20. Page 10, line 10: together with N-WASP or the Arp2/3 complex

21. Page 10, line 20: (see a model in Fig. 4i). Remove "a"

22. Page 29, line 15: and stained with for

23. Page 29, line 8: sequences (plural)

24. Page 37, line 19-20: sentence in italic

25. Page 41, line 20: Polyclonal antibodies were generated against MirA in a rabbit host. Is "for both organisms" correct?

26. Page 54, line 5: vs Δ mirA (singular, only one delta strain is shown in the graph)

27. Page 56, line 9: Please check if the sentence is correct. "Quantification of the percent of Δ mirA M. marinum co-localizing with either MirA..."

Figure 1:

28. Panel e: The small zoomed images are not labelled and therefore not implicit. To make it easier understandable, please add the rectangular indicating N-WASP positive bacteria also to the merged images.

29. Panel e: The N-WASP signal in the merged image of the delta3581 appears quite dim. Please check if the image intensity is the same than in the other merged images of this panel.

30. Panel g: In both images, bacteria were cut perpendicular and only a small part of the bacteria is shown. The other parts might be still inside the vacuole in other slices. Please exchange these images or move them to the supplementary material. As mentioned above, please verify phagosome escape of the Δ mirA mutant with another method.

Figure2:

31. Panels b and c: Please add the rectangular indicating the position of the zoom also to the merged images.

Figure3:

32. Panel a: Please add the rectangular indicating the position of the zoom also to the MirA images.

33. Panel b: Actin-based motility of host LDs. The corresponding movie (ext. data video 1) is quite striking, however, the snapshots from the movie do not reflect this. Please revise and change to images with labels.

34. Panel d: Please add the rectangular indicating the position of the zoom also to the MirA images.

35. Panels h and i: Please add the rectangular indicating the position of the zoom also to the merged images.

REVIEWER COMMENTS

Reviewer #1 (Remarks to the Author):

“The manuscript by Norbert S. Hill & Matthew D. Welch titled “A mycobacterial glycine-rich protein governs actin-based motility” is an original work that points the identification and characterization of MirA an M. marinum actin-based motility factor, a member of the PE_PGRS family of glycine-rich of ESX-5-secreted proteins. The novelty of this work relies on:

(1) The fact that MirA while binding directly to WASP/NWASP to recruit the actin nucleating factor ARP2/3 similar to that described in Shigella flexneri IcsA, enterohemorrhagic Escherichia coli EspFU, and Chlamydia trachomatis TmeA it does not target the GTPase binding domain indicating a distinct role of actin nucleator or actin promotor. Instead MirA targets a small positively charged basic region only known to interact with a phosphoinositol lipid (PIP2) with potential to represent a new class of microbial and host actin regulator.

(2) MirA uses an amphipathic helix to anchor into the mycobacterial outer membrane and, also the surface of host lipid droplet organelles indicating a relevant virulent factor in manipulating the host lipid droplets and potential associated vesicles intracellular trafficking representing a model to understand how the PE_PGRS proteins contribute to mycobacterial pathogenesis.”

Response: We thank the reviewer for recognizing these various contributions of our work and for their time to construct a thoughtful response.

“While the gene product MMAR_3581 looks to be encoded only by M. marinum among mycobacterial species with a truncated form encoded by the closely related pathogen Mycobacterium ulcerans it cannot be excluded the possibility to similar genes operating in M. tuberculosis.

As the authors stated the PE_PGRS family of proteins at the level of individual contribution to infection is poorly understood due to lack of sequence homology.

According to KEGG orthology database there's a good match between the MMAR-3581 and the Mtb Erdman strain mtn: ERDMAN_3853 ortholog while no actin based motility was demonstrated so far on virulent Mtb escaping to the cytosol in any model used of host cells ([https://www.kegg.jp/ssdb-bin/ssdb_best?org_gene=mmi:MMAR_3581; GenBank: BAL67626.1](https://www.kegg.jp/ssdb-bin/ssdb_best?org_gene=mmi:MMAR_3581;GenBank: BAL67626.1)). It would be interesting to assess the effects of the ortholog gene product of the Erdman strain of Mtb in a context were actin tail formation is facilitated the ortholog protein will also display a similar phenotype.

The impact of the present work is that this family of proteins are related with important virulence factors of pathogenic mycobacteria: a similar system in Mtb will help bacteria spread to new non infected cells and in addition the ability to induce actin tails in host lipid droplets may be involved in host lipid droplets trafficking and signaling for host manipulation while promoting bacterial survival.”

Response: We appreciate the effort the reviewer spent bioinformatically probing for a MirA homolog in *M. tuberculosis*. The candidate homolog, ERDMAN_3853 (truncated homolog in *M. tuberculosis* strain H37Rv PE_PGRS56), does show 54% sequence identity with MirA. However, because of the glycine-rich sequence of PE_PGRS proteins, MirA has 40-55% sequence identity to many of the PE_PGRS proteins. It has been an outstanding question in the mycobacterial field as to what regions of the PGRS sequence are important to PE_PGRS function. Without knowing this, it is not possible to accurately assign true homologs among this large protein family without a higher degree of sequence identity.

As the reviewer correctly states, robust actin polymerization has not been observed at the *M. tuberculosis* bacterial surface. Thus, we anticipated that the candidate homolog ERDMAN_3853 would not activate host N-WASP to stimulate actin polymerization. Nonetheless, to satisfy this curiosity, we expressed this protein in U2OS cells and examined for F-actin. Ectopically expressed ERDMAN_3853 localized to lipid droplets (ERDMAN_3853 has a predicted amphipathic helix (hydrophobic moment score = 0.423): $_{97}\text{QNLLNLI}\text{NAPTQTLLGRP}_{114}$) but did not induce actin filament assembly at the lipid droplets surface (shown below). Because of the negative result, we have opted not to include this in our revised manuscript.

“The work here shown is original and sound. The work is well supported with methods and full demonstration of the enrolment of MirA in recruitment of WASP/NWASP and interacting proteins either to the surface of the bacteria either to host lipid membrane droplets. The experiments are robust and well controlled with lots of supporting experiments

Curiously the authors demonstrate that MirA targets a charged region of the phospholipid layer that interacts with PIP2. While it is described the interaction of WASP with PIP2 (PMID: 23445807) there are other ABP described to serve as intermediated in

the recruitment of actin via PIP2 to host vesicle membranes to induce the formation of actin filaments. The ERM family of proteins were previously shown relevant to induce the formation of actin tracks from phagosomes containing latex beads (PMID: 10637224) and similar system was suggested to occurs in phagosomes containing non-pathogenic and pathogenic mycobacteria contributing to the pathogenesis of Mtb and M. avium (PMID: 12942085). While this event is related with the phagosomal context and membranes maybe this should be explored in the discussion for a possible relationship between the actin associated membranes and PIP2, ERM and a possible new role of MirA in promoting actin filamentation.

The work is robust and suited for publication taken in consideration a few points outlined to introduce in the discussion.”

Response: The reviewer has made an interesting connection between the MirA-N-WASP interaction and the ERM proteins (Ezrin, Radixin, and Moesin) that link actin filaments to the mycobacterial containing vacuole. Relatedly, after phagocytosis, the actin nucleation promoting factor WASH (a WASP-family protein) associates and generates F-actin on the *M. marinum* vacuole (PMID: 24119059). It is possible that the ERM proteins observed by Anes et al., 2003 (PMID: 12942085) are associated with the actin filaments generated by WASH. However, we do not hypothesize there would be a strong link between MirA-stimulated actin nucleation and ERM protein actin filament binding and therefore have not included mention of this in our short discussion section.

As we wrote in our Discussion section, it has been as curiosity as to why the basic domain of WASP/N-WASP is necessary to stimulate *M. marinum* actin-based motility. This domain of N-WASP is known to interact with the signaling lipid PIP₂ to direct actin polymerization at cellular membranes. Because phosphoinositol lipids are a significant component of mycobacterial membranes, we had initially hypothesized that *M. marinum* encoded or recruited a kinase to create PIP₂ on its surface to recruit N-WASP. However, we (unpublished data) and previous researchers (PMID: 16199520) failed to detect PIP₂ on the bacterial surface using multiple approaches. Without PIP₂ it is unlikely that the ERM proteins would be recruited to the *M. marinum* surface to play a role in the organization of MirA-stimulated actin filaments. Additionally, had an ERM protein been identified through our affinity purification-mass spectrometry approach (Fig. 6 and Supplementary Data 2), we would have opted to incorporate discussion of the seminal works regarding actin-coated phagosomes.

Additionally, the actin polymerization in these two situations play entirely opposite roles in bacterial pathogenesis/host response. The actin around phagosomes observed by Anes et al. 2003 (PMID: 12942085) is associated with bacterial killing. Live *M. tuberculosis* or *M. avium* dampen this actin assembly to, perhaps, avoid trafficking through lysosomal pathway and subsequent degradation. In other words, the actin polymerization around mycobacterial vacuoles in Anes et al. 2003 is an anti-bacterial mechanism. The actin generated by MirA serves an entirely different role as it is derived from a bacterial protein and part of a mechanism promoting bacterial cell-to-cell spread.

Reviewer #2 (Remarks to the Author):

“In this study, Hill and Welch identify a M. marinum protein (MirA) that promotes actin polymerization in infected cells. Starting from a M. marinum mutant library the authors identify the gene MMAR_3581 as a gene required for proper M. marinum spread in infected cells. Designing very elegant experiments the authors demonstrate that MirA is necessary to activate actin polymerization by interacting with the N-WASP and the Arp2/3 complex. The results of this study shed new light on the mechanisms used by M. marinum to spread in infected cells and most importantly implicate a member of a family of important mycobacterial proteins (PE_PGRS) in the pathogenesis of mycobacterial infections.”

Response: We thank the reviewer for recognizing these various contributions of our work and for their time to construct a thoughtful response.

Major comments:

“The authors conclude that the amphipathic linker AH region is necessary and sufficient for localization of MirA on the bacterial surface. However, there are few issues that in my opinion shall be clarified before reaching this conclusion.

This conclusion implies that the PE domain is cleaved off from the full length MirA protein, with cleavage occurring immediately after the PE domain (somewhere between position 88-101). However, the data provided to support cleavage at this site are weak, since the WB in extended Fig 3A indicate that most of the protein is intact and not cleaved. These results are important considering previous studies indicating that, in M. marinum and in Mtb (Daleke 2011, Burgraaf 2020, De Maio 2021), the cleavage site of the PE domain in PE_PGRS proteins is downstream of what indicated in this study.”

Response: We thank the reviewer for this comment and for their expertise regarding PE_PGRS proteins. The reviewer is correct that our assessment of MirA cleavage was over-stated as we did not provide evidence as to where MirA is proteolytically processed after secretion (e.g., by Edman degradation, mutational analysis, etc.). This would be experimentally possible but challenging since MirA is expressed/secreted only during infection and not in broth conditions. As such, we have removed any mention of a specific MirA cleavage site. Additionally, we have examined MirA's subcellular fractionation profile (discussed below), which demonstrates MirA is proteolytically processed upon secretion like other known PE_PGRS proteins.

“These results are important considering previous studies indicating that, in M. marinum and in Mtb (Daleke 2011, Burgraaf 2020, De Maio 2021), the cleavage site of the PE domain in PE_PGRS proteins is downstream of what indicated in this study.”

Response: We recognize the reviewers concern as it has also been of interest to us based on limited knowledge of PE_PGRS secretion. The only defined example of PE_PGRS proteolytic processing is the outer membrane lipase LipY (PMID: 21471225;

31662454). As the reviewer is likely aware, LipY is categorized as a PE_PGRS protein, but it is an atypical PE_PGRS protein as it lacks the canonical glycine-rich PGRS sequence. And, because there are few studies in this area, it is unclear whether LipY processing is a singular or representative case for the PE_PGRS protein family. Nonetheless, the studies of LipY are of excellent quality and the most detailed examination of PE_PGRS protein secretion and processing.

To articulate the concern as we understand it, three PecA proteolytic sites have been identified in LipY (PMID: 31662454). One upstream and two downstream of the best candidate amphipathic helix we determined using HeliQuest (PMID: 18662927) (diagramed below). If all three sites were cleaved immediately after secretion, only a portion of the linker domain would be anchored into the bacterial membrane by the amphipathic helix. Importantly, the PGRS domain would not be attached to the amphipathic helix and would have to use a different bacterial outer membrane targeting mechanism. To reconcile these data, one could speculate that PecA does not cleave all three sites within LipY immediately upon secretion, but sequentially over time. For instance, an initial cleavage at the furthest N-terminal site would cleave off the PE domain but leave the amphipathic helix to tether the PGRS domain to the outer membrane. At a later point, PecA could cleave the other sites to promote protein turnover at the bacterial surface.

Regardless of why PecA has three cleavage sites in LipY, we present multiple lines of evidence that MirA's amphipathic helix is the primary feature that tethers MirA to the mycobacterial outer membrane. First, ectopically expressing just MirA's amphipathic helix fused to tdTomato is sufficient for localization to the surface of host lipid droplets (Fig. 5b) and the mycobacterial surface (this data has been added during the revision process as Fig. 5c and is shown below). This leaves little ambiguity that this feature is sufficient for insertion into these surfaces.

Second, ectopically expressed full-length MirA in multiple eukaryotic cell types results in localization to both lipid droplets and the bacterial surface. This demonstrates that MirA bacterial membrane insertion is not coupled to secretion and is consistent with how proteins with amphipathic helices target membranes. Crucially, ectopically expressing a MirA Δ AH mutant, which is expressed at the same level as WT (Supplementary Fig. 6b), does not localize to either surface (Fig. 5d, e, g, and Supplementary Fig. 7a). This demonstrates the amphipathic helix is necessary for membrane tethering.

Third, the MirA Δ AH mutant endogenously expressed under its native promoter during infection, which has no appreciable decrease stability (Supplementary Fig. 6a, c), is not detected at the bacterial surface by immunofluorescence microscopy (Fig. 5f).

Supporting the absence of MirA Δ AH at the surface is the paucity of actin filaments at the bacterial surface. Instead, we observe punctate MirA foci in the host cytoplasm. All together, these data suggest the MirA amphipathic helix is sufficient and necessary for membrane targeting.

During the review process, we also probed for wild-type MirA and MirA Δ AH in the genapol-treated bacterial pellet, genapol supernatants (thought to be components extracted from the bacterial surface), and the host cytosol by subcellular fractionation using bacterial cells from infection of Raw 264.7 cells at 48 hpi (Supplementary Fig. 6c and shown below). We observed MirA Δ AH in all three fractions, including the genapol-supernatant suggesting it is associated with the bacterial surface. This data is in conflict with our observations described above. We speculate that MirA Δ AH is localized to a genapol-extractable region of the bacterial envelope but not correctly oriented in the outer leaflet without the amphipathic helix. The result nevertheless raises the possibility of additional membrane targeting mechanisms for MirA (and the PE_PGRS family). We have incorporated the results from the subcellular fractionation into the manuscript (page 9, line 21; Supplementary Fig. 6c).

“Additionally, we probed for MirA by subcellular fractionation and, surprisingly, detected wild-type levels of MirA Δ AH in the genapol supernatant (Supplementary Fig. 6c). This suggests MirA Δ AH localizes and is retained in the bacterial outer envelope. Since we do not observe surface-associated MirA or actin polymerization by microscopy, this may represent a MirA fraction that is not fully secreted to the surface or correctly oriented in the outer leaflet to recruit the actin machinery. Alternatively, this data could suggest MirA has additional surface localization mechanisms beyond the amphipathic helix.”

Supplementary Fig. 6c:

“However, the data provided to support cleavage at this site are weak, since the WB in extended Fig 3A indicate that most of the protein is intact and not cleaved.”

Response: We appreciate the reviewer’s attention to detail and recognize that our initial submission poorly articulated the differences in our MirA immunoblotting.

In Fig. 3a, we probed for secreted MirA to determine if there was a correlation with the timing of actin-based motility. In this experiment, we show wild type genapol supernatants from *M. marinum* isolated from infected Raw 264.7 host cells at various time points. We have made this clear in the text (page 6, line 16; page 27, line 14).

Fig. 2a (genapol supernatants):

In Supplementary Fig. 6a, we examined MirA in bacterial whole cell lysates from an infection at 48 hpi. This includes the MirA population in the bacterial cytosol and bacterial envelope. We have noted this in the manuscript (page 27, line 14 and Supplementary Document page 11, line 2).

Supplementary Fig. 6a (whole cell lysate):

We observe a predominant larger native form of MirA (black arrowhead), and a smaller band (white arrowhead) corresponding to the loss of the PE domain (~8 kDa). If MirA is similar to other PE_PGRS proteins (e.g., 33757409, etc.), then the cleaved form would represent MirA that has been secreted to the surface. To this, we employed subcellular fractionation (shown above and Supplementary Fig. 6c). Important to your concern, we observe a full-length MirA in the bacterial cytoplasm (genapol-treated pellet) and the processed form of MirA in the genapol supernatant (outer bacterial envelope). This size differential corresponds to a loss of the PE domain (~8 kDa) suggesting that MirA is processed upon secretion like known PE_PGRS proteins.

We additionally observe other previously unobserved MirA bands in the various fractions. 1) An additional MirA species in the bacterial cytosol not observed in whole cell lysates. We hypothesize this is a MirA degradation product that occurs because of the processing lag between host cell lysis, genapol extraction, and bacterial cell lysis. 2) We observe multiple MirA species in the genapol supernatant: full length and two processed versions. This suggests MirA is proteolytically cleaved at least twice, perhaps similar to LipY. 3) We observe that a portion of unprocessed MirA is lost from the bacterial surface into the host cytosol.

*“It is also important to clarify whether cleavage of the PE domain is necessary to allow the MirA amphipathic AH domain proper localization on mycobacterial membrane and lipid droplets. A better characterization of this aspect is needed to support the hypothesis: comparison between the MirA expressed in *M. marinum* and that ectopically expressed in transfected cells may be useful to clarify this issue.”*

Response: Our results suggest cleavage of the PE domain plays no role in the ability of the amphipathic helix to insert into the mycobacterial or lipid droplet surface. We observe that both ectopically expressed full-length or MirA Δ PE (Δ 1-87) localize to both lipid droplet and *M. marinum* surfaces by immunofluorescent microscopy with no discernable difference (lipid droplets WT (Fig. 3g and Fig. 4b) and Δ PE (Fig. 4d-f, 6a, Movie 3), and *M. marinum* WT (Fig. 3g) and Δ PE (Fig. S7). Further, we do not observe any differences in actin polymerization, thus the PE domain does not appear to impact binding to either membrane or interaction with N-WASP *in vivo*.

“Moreover, subcellular fractionation may provide insights on the cleavage of the PE domain and stability of the MirDeltaAH mutant (see comment below).”

From what indicated in the methods, the MirDeltaAH functional deletion mutant was generated by omitting the sequence coding for the linker region (residues 102-119). It would be interesting to know whether the lack of this linker domain in the mutant affect protein stability and secretion through the mycobacterial membrane. This is an important domain and direct fusion between the PE domain and the region upstream the AH linker may affect the protein functional properties.”

Response: Thank you for the comment. We show the Mir Δ AH mutant is both stable and is secreted (Supplementary Fig. 6). MirA mutants lacking either the PE domain or the amphipathic helix have similar stability to wild-type MirA expressed ectopically in the U2OS cell line (Supplementary Fig. 6b). Additionally, complementing a Δ mirA strain with a chromosomally integrated P_{mirA} -mirA Δ AH-flag exhibits similar expression levels and proteolytic processing in whole cell lysates (now Supplementary Fig. 6a). Further, the subcellular localization of the Δ AH mutant has normal expression bacterial cytosol and is secreted to the bacterial surface and beyond (Supplementary Fig. 6c and shown above). We have made note that this mutant is both stable and secreted as to avoid confusion (page 9, line 15):

“Likewise, in spite of normal expression and secretion (Supplementary Fig. 6a, c) ...”

Minor comments:

“Page 6, line 6: Rather than using the term “PE family domain” I suggest “PE domain”.”

Response: We have changed “PE family domain” to “PE domain” in all cases of our manuscript (page 7, line 11, 17; page 9, line 7; page 31, line 8).

“Page 7, line 18: PE domain rather than PE family; moreover, PGRS domain rather than the plural domains.”

“Page 8, line 4: probably endogenously rather than endogenous.”

Response: Thank you for your close reading and we have made the suggested change (page 9, line 14).

“Page 10, line 16: “... MirA is the M. marinum actin-based motility factor”. It may be more appropriate to state that “... MirA is a M. marinum actin-based motility factor”

Response: Your point is taken, but there is no evidence that *M. marinum* possesses multiple actin-based motility factors. We have never observed actin comet tails at the surface of Δ mirA mutants after extensive characterization. Changing the text to, “MirA is a *M. marinum* actin-based motility effector...”, then confusingly infers there is evidence for multiple actin-based motility factors in *M. marinum*.

“Page 37, line 20: “... PE super domain..”. PE domain.”

Reviewer #3 (Remarks to the Author):

“Hill & Welch make substantial efforts to investigate the molecular mechanisms underlying actin polymerization at the surface of Mycobacterium marinum to promote intracellular mobility and dissemination to neighboring cells during infection. To this end, M. marinum recruits the host proteins WASP and/or N-WASP to stimulate actin nucleation by the Arp2/3 complex.

The main findings are that (i) M. marinum secretes the factor MirA, a member of the PE_PGRS family of Esx-5 secreted proteins, to recruit and activate N-WASP and that (ii) MirA is anchored to both, the surface of M. marinum and host lipid droplets (LDs) via its amphipathic helix to promote bacterial but also LD actin-based motility. The findings are new and will be of significance for the field since only few PE_PGRS proteins have been described to be involved in host-pathogen interactions, however the molecular mechanism remain, in contrast to the here characterized MirA, elusive. The study meets high quality standards. The microscopy images and videos, in particular, are of exceptional quality. I have mostly minor comments.”

Response: We thank the reviewer for recognizing these various contributions of our work and their time to construct a thoughtful response and thorough proofreading.

B. Major comments

1. *“To rule out that the defect in actin-based motility of the delta_mirA mutant was not due to a problem of cytosolic access, the authors carried out an experiment that is based on electron microscopy (Fig. 1g, h). Because quantifying dynamic processes like phagosomal escape using 2D electron microscopy is challenging and also the escape kinetics of M. marinum in U20S cells are not known the authors should analyze phagosome escape also with another method (for example co-localization with ubiquitin or xenophagy markers) and include more time points. Only one late time point was chosen (60 hpi), which makes it incomparable to other experiments (examples: Fig. 1b, 1e -> 48 hpi).”*

Response: We thank the reviewer for this comment and we appreciate proposing an experiment to address the concern. We hypothesize that MirA does not have a role in escape from the mycobacterial containing vacuole (MCV) based on the observation that *M. marinum* initiates escape from the MCV into the host cytoplasm by 2-4 hpi after invasion (Fig. 2d and PMID: 19436699; 21447143), yet we do not detect secreted MirA until ~8 hpi (Fig. 3a).

As the reviewer suggested, we examined polyubiquitin staining between WT and the Δ *mirA* mutant in primary mouse macrophages at 4, 24, or 48 hpi. We observed similar frequency of ubiquitin staining with *M. marinum* as previous studies (PMID: 19436699;

21447143). Importantly, we did not detect a difference in polyubiquitin staining between WT and $\Delta mirA$, supporting that the $\Delta mirA$ mutant does not have a severe MCV escape defect. This further suggests that MirA does not play a significant role in autophagy evasion. This data has been added to Supplementary Fig. 5c, d as shown below.

Additionally, we used internalized CM-Dil dye staining to visualize the *M. marinum* vacuolar membrane by fluorescent microscopy as previously described (PMID: 14597736; 18852239) and outlined in our Methods section (page 23, line 20). We examined *M. marinum* colocalization with vacuolar staining over the course of infection (4, 24, 48, and 72 hpi) in three independent experiments. This approach also supported that the fraction of $\Delta mirA$ bacteria in the cytosol equals or exceeds that of wild-type bacteria, and the lack of actin polymerization is not a consequence of defective vacuolar escape. This more comprehensive examination has replaced our TEM data in Fig. 2c, d (shown below), which is now in Supplementary Fig. 5a, b.

Furthermore, the following statement has been added to the Results section (page 6, line 2):

“Further, the Δ MMAR_3581 bacteria were not deficient in actin-based motility due to a defect in escaping from the vacuole into the host cytoplasm. Δ MMAR_3581 bacteria were observed in the host cytosol with same or higher frequency as wild type when either staining the vacuolar membrane (Fig. 2c, d) or by transmission electron microscopy (Supplementary Fig. 5a, b). Additionally, Δ MMAR_3581 had a similar frequency of polyubiquitin colocalization compared with wild type, which serves as an indirect marker of vacuolar integrity (Supplementary Fig. 5c, d)^{13,14}. This also infers that MMAR_3581 may not play a role in autophagy evasion like other bacterial actin-based motility effector proteins^{15–17}.”

2. “Page 4, lines 12,13: bacteria were assessed for association with actin clouds, actin tails, and N-WASP over the course of infection. Please explain how actin clouds/actin tails/N-WASP were defined and quantitated (manually or automatically?)”

Response: We have clarified this point in the Methods (page 21, line 19):

“Colocalization was manually scored using the ImageJ cell counter plugin with an actin structure at the bacterial $>1 \mu\text{m}$ counted as an actin tail. For Fig. 1e, f, wild type (BHM103), Δ mirA (BHM144), and Δ mirA + P_{3581} -3581 (BHM161) were used and 50-400 bacteria/replicate were calculated over three biological replicates.”

3. “To be included in the discussion: MirA appears to be mainly enriched at one bacterial pole but completely surrounds host LDs. Please include a possible explanation (lipid composition, membrane curvature...) also in the context that actin polymerization takes place at one pole of *M. marinum* and at one concentrated spot on the LD surface.”

Response: The reviewer raises an interesting point. It is possible that MirA has a predilection for positive membrane curvature or a particular lipid composition, but the primary reason MirA uniformly surrounds lipid droplets is simply due to high MirA expression levels. Ectopically expressing MirA in eukaryotic cells creates a situation in which MirA saturates membrane insertion sites. This occurs on the lipid droplet (Fig. 4a), but also the *M. marinum* surface (Fig. 2g and Supplementary Fig. 7b). Complementing for *M. marinum* actin tails (not actin clouds) *in trans* (as shown in Fig. 2h) only occurs when there is a reduced MirA: *M. marinum* ratio. Further, we are currently exploring how MirA primarily localizes to a single pole during infection.

4. “Page 7, line 17: Proline is thought to destabilize amphipathic helices. Fig. 3f. shows the putative amphipathic helix of MirA including a proline. Please include in the results section or the discussion that this proline might disrupt the helix.”

Response: We also have been interested about the “helix-busting” residue in the MirA amphipathic helix. Curiously, the proline residue is a conserved feature within the hydrophobic face of putative PE_PGRS amphipathic helices as our analysis shows in Supplementary Data 1. This may suggest that the proline residue is important to membrane targeting. Interestingly, there are other examples of amphipathic helices with

a proline residue that target positive curvature, such as the SpoVM from *Bacillus subtilis* (PMID: 12562810).

5. “*Perilipins are LD proteins that have mainly protective function (to shield the hydrophobic LD core from cytosolic lipases) and are targeted to the phospholipid monolayer of LDs via an amphipathic helix. The role of MirA on the LD surface remains speculative; therefore, it would be interesting to know if the number of host LDs is changed upon MirA expression. Does MirA expression induce the formation of LDs?*”

Response: In our hands, ectopic expression of MirA in host cells reduces the localization PLIN2 (a canonical lipid droplet protein) on the lipid droplet surface (data not shown). This suggests that MirA occludes normal host lipid droplet proteins from binding and could impact normal lipid droplet cellular dynamics. Thus, we expect the biology of host lipid droplets to be altered when MirA is expressed. As a cursory examination, we assessed the size and number of lipid droplets \pm *mirA* expression as a possible indicator that MirA affects lipid droplet biogenesis, fusion, or disassociation. However, we found no significant effect on these parameters in cells expressing MirA. We hypothesize that saturating lipid droplets with a foreign protein likely impacts lipid droplet function, but may be more obvious in other situations (e.g. lipotoxic stress). We have included this data to Supplementary Fig. 8a, b (shown below) and Results section (page 8, line 1):

“Expression of MirA did not affect lipid droplet size or number (Supplementary Fig. 8a, b).”

6. “*Concerning the actin polymerization assays: by comparing both figures (Fig. 4f, 4g) and by following the text it is difficult to understand the differences between both experiments. The authors should change the figure legends to make it more implicit. Only one representative experiment is shown (for both experiments in Fig. 4f, 4g), therefore the authors should include a second representative in the supplementary data.*”

Response: We agree that the figure legend could have been written with more clarity. We have revised it as follows (page 60, line 9):

“**d** Pyrene actin (1 μM , 10% labeled) polymerization reactions with either MirA (200 nM), MirA and the Arp2/3 complex (50 nM), or MirA and N-WASP (200 nM). **e** Pyrene actin

(1 μ M, 10% labeled) polymerization reactions with the Arp2/3 complex (50 nM), N-WASP (200 nM), and increasing MirA concentrations."

As requested, we have also included an auxiliary representative of the pyrene-labeled actin polymerization curve in Supplementary Fig. 10a, c.

7. "Discussion: Page 12, line 10: In the manuscript it was not formally shown that the MirA PGRS domain directly binds and activates N-WASP. Therefore, the authors should formulate the sentence more cautiously."

Response: The reviewer is correct that we do not show that the PGRS domain activates N-WASP and have edited the text to remove those statements.

8. "Page 12, lines 19-21: "However, the GBD of WASP/N-WASP is dispensable for *M. marinum* 20 actin-based motility, and instead MirA targets a small positively charged basic region only known to interact with a phosphoinositol lipid (PIP₂)" As this was not directly demonstrated here, the authors should formulate this more cautiously."

Response: The reviewer is correct. We have amended the sentence to read (Page 15, line 20):

"Curiously, only the eight amino acid lysine-rich basic region of WASP/N-WASP, not the GBD, is necessary for *M. marinum* to stimulate actin-based motility⁹. This domain binds the phosphoinositol lipid PIP₂ to coordinate actin polymerization at cellular membranes^{54,55}. Why this domain is uniquely important to *M. marinum* among pathogens that target WASP/N-WASP is unclear, but the discovery of MirA may provide a new model to understand how WASP family proteins can be regulated."

C. Minor comments

9. "U2OS are not commonly used for infections. Please explain why U2OS cells were used in this study."

Response: U2OS cells are indeed an atypical choice for mycobacterial infection. We required a cell line that formed flat, confluent host cell monolayers that allow for easier visual assessment of bacterial spread by actin-based motility. We have included this rationale in the text (page 20, line 3):

"The osteoclast cell line U2OS was used in some experiments for two reasons: they form flat, stationary, and confluent monolayers that allow for easy assessment of cell-to-cell spread; and they are non-hematopoietic cells that express N-WASP but not WASP."

10. "Wild-type or wild type, please decide for one and check throughout"

Response: Our understanding is that "wild-type" is an adjective and "wild type" refers to the noun. We have tried to fix any errors in the text.

11. *“To facilitate reading. In panels with microscopy images, please indicate the times post infection.”*

Response: The time post-infection details have been added to all microscopy panels.

12. *“It is not clear if LDs were induced prior to the LD experiments in various cells types (A549, U2O2, HEK293), please mention briefly in M&M or in the figure legends.”*

Response: Lipid droplet synthesis was not induced by adding oleic acid in our experiments except for when examining lipid droplet cell-to-cell spread (Fig. 4f). We have added a sentence to the Method section highlighting this fact (page 18, line 4; page 31, line 2).

13. *“Page 1, line 21: MirA is an abbreviation, please write the full name once in the abstract”*

Response: We have fixed this oversight (Page 2, line 8).

14. *“Page 4, line 10: remove comma after e.g.”*

Response: Our understanding is that a comma is appropriate after “e.g.” if multiple items then follow. We have double-checked our text to correct any errors.

15. *“Page 5, line 8: 2x that, delete one”*

Response: Thank you for your close reading. We have fixed this error.

16. *“Page 5, line 19: apparently here “Percent of MirA-positive bacteria colocalizing with N-WASP or an actin tail “was measured. Please change the text of the manuscript accordingly and remove “that colocalized with MirA”.”*

Response: We have rewritten the sentence to (Page 6, line 23):
“At 72 hpi, the percent of MirA-positive bacteria with surface-associated N-WASP or an actin tail was 71% or 63%, respectively.”

17. *“Page 6, line 16: Fig. 2h, white arrow (not 2g)”*

Response: Thank you for catching this typo (corrected on page 7, line 22).

18. *“Page 8, L. 13: MirA uses an amphipathic alpha helix to insert in the phospholipid monolayer of both host LDs and the mycobacterial outer membrane. The authors should be cautious as the mycobacterial outer membrane composition is quite complex. I would recommend to write that it might resemble a monolayer.”*

Response: We have removed any statement that the mycobacterial surface is a phospholipid monolayer from our manuscript.

19. "Page 9, lines 8 and 11: (*This approach also identified....*) Remove parenthesis."

Response: The parentheses have been removed from this sentence (Page 11, line 3).

20. "Page 10, line 10: *together with N-WASP or the Arp2/3 complex*"

Response: "Either" has been removed from this sentence (Page 12, line 6).

21. "Page 10, line 20: (*see a model in Fig. 4i*). Remove "a""

Response: The "a" has been removed (Page 13, line 6).

22. "Page 29, line 15: *and stained with for*"

Response: The "with" has been removed (Page 19, line 18).

23. "Page 29, line 8: *sequences (plural)*"

Response: We have modified the sentence to (Page 17, line 8):
"The full sequences of plasmids used in this study are provided in Supplementary Table 3a."

24. "Page 37, line 19-20: *sentence in italic*"

Response: Thank you for noticing this error and it has been rectified.

25. "Page 41, line 20: *Polyclonal antibodies were generated against MirA in a rabbit host. Is "for both organisms" correct?*"

Response: Thank you for the close reading. We have removed "for both organisms" as only anti-MirA raised in rabbit was used in this study (Page 35, line 5).

26. "Page 54, line 5: *vs Δ mirA (singular, only one delta strain is shown in the graph)*"

Response: Now reads (Supplementary document - Page 5, line 3):
"b Growth curve of wild type versus Δ 3581 mutant bacteria during infection of primary mouse macrophage (BMDM) cells."

27. "Page 56, line 9: *Please check if the sentence is correct. "Quantification of the percent of Δ mirA M. marinum co-localizing with either MirA..."*"

Response: We have attempted to make the sentence more lucid. It now reads (Supplementary document - Page 15, line 2):

“a Percent of Δ mirA bacteria colocalized with either MirA, an actin cloud, or an actin comet tail in host cells ectopically expressing either MirA, MirA ^{Δ PE}, MirA ^{Δ AH}, or an empty vector control.”

“Figure 1:”

28. “Panel e: The small zoomed images are not labelled and therefore not implicit. To make it easier understandable, please add the rectangular indicating N-WASP positive bacteria also to the merged images.”

Response: The rectangular outlines highlighting the zoomed images have been moved to the merged panel to improve the Fig. 2a.

29. “Panel e: The N-WASP signal in the merged image of the delta3581 appears quite dim. Please check if the image intensity is the same than in the other merged images of this panel.”

Response: Thank you for pointing this out. We have adjusted the intensity level to equivalent levels in the merged panel in Fig. 2a.

30. “Panel g: In both images, bacteria were cut perpendicular and only a small part of the bacteria is shown. The other parts might be still inside the vacuole in other slices. Please exchange these images or move them to the supplementary material. As mentioned above, please verify phagosome escape of the Δ mirA mutant with another method.”

Response: As previously mentioned, we have added two additional pieces of data that Δ mirA bacteria are able to access the host cytosol with similar kinetics. Data examining the colocalization of bacteria with internalized CM-Dil staining (a proxy of the mycobacterial containing vacuole) is now in Fig. 2c, d. We have moved the transmission electron microscopy data to Supplementary Fig. 7 along with the frequencies of polyubiquitin localization between WT and Δ mirA bacteria.

“Figure 2:”

31. “Panels b and c: Please add the rectangular indicating the position of the zoom also to the merged images.”

Response: The rectangular boxes have been moved to the merged panel to improve Fig. 3b, c.

“Figure 3:”

32. “Panel a: Please add the rectangular indicating the position of the zoom also to the MirA images.”

Response: Boxes have been added to the MirA images in Fig. 4a.

33. “Panel b: Actin-based motility of host LDs. The corresponding movie (ext. data video

1) is quite striking, however, the snapshots from the movie do not reflect this. Please revise and change to images with labels.”

Response: We have changed this figure better capture the dynamic nature of the MirA-induced lipid droplet actin rocketing. We incorporated a figure with better snapshots (Fig. 4b) and cognate movie (Movie 2) with object tracking to show the trajectory of the lipid droplets. Additionally, we added object tracking to Movies 1, 2, and 5 to better capture the movement of the MirA-coated lipid droplets or polystyrene beads.

34. “Panel d: Please add the rectangular indicating the position of the zoom also to the MirA images.”

Response: Boxes have been added to the images to improve Fig. 4f.

35. “Panels h and i: Please add the rectangular indicating the position of the zoom also to the merged images.”

Response: We have re-cropped these images rendering zoom highlights unnecessary (Fig. 5d-f).

Reviewers' Comments:

Reviewer #1:

Remarks to the Author:

The authors responded to all concerns of my assessment.

I support the publication of the manuscript.

Reviewer #2:

Remarks to the Author:

[No further comments for author]

Reviewer #3:

Remarks to the Author:

The paper on the characterization of MirA, an *M. marinum* actin-based motility factor that interacts with WASP/NWASP to recruit the actin nucleating factor ARP2/3, is a really beautiful piece of work.

All experiments have been performed with precision and exquisite attention to detail. All my previous comments and concerns were addressed within the revision process. Therefore I highly recommend the paper for publication in Nature Communications.